# Pessimistic Nonlinear Least-Squares Value Iteration for Offline Reinforcement Learning

## Abstract

Offline reinforcement learning, where the agent aims to learn the optimal policy based on the data collected by a behavior policy, has attracted increasing attention in recent years. While offline RL with linear function approximation has been extensively studied with optimal results achieved under certain assumptions, the theoretical understanding of offline RL with non-linear function approximation is still limited. Specifically, most existing works on offline RL with non-linear function approximation either have a poor dependency on the function class complexity or require an inefficient planning phase. In this paper, we propose an oracle-efficient algorithm PNLSVI for offline RL with non-linear function approximation. Our algorithmic design comprises three innovative components: (1) a variance-based weighted regression scheme that can be applied to a wide range of function classes, (2) a subroutine for variance estimation, and (3) a planning phase that utilizes a pessimistic value iteration approach. Our algorithm enjoys a regret bound that has a tight dependency on the function class complexity and achieves minimax optimal problem-dependent regret when specialized to linear function approximation. Our theoretical analysis introduces a new coverage assumption for nonlinear Q function, bridging the minimum-eigenvalue assumption and the uncertainty measure widely used in online nonlinear RL. To the best of our knowledge, this is the first statistically optimal algorithm for nonlinear offline RL.

## 1 Introduction

Offline reinforcement learning (RL), also known as batch RL, is a learning paradigm where an agent learns to make decisions based on a set of pre-collected data, instead of interacting with the environment in real-time like online RL. The goal of offline RL is to learn a policy that performs well in a given task, based on historical data that was collected from an unknown environment. Recent years have witnessed significant progress in developing offline RL algorithms that can leverage large amounts of data to learn effective policies. These algorithms often incorporate powerful function approximation techniques, such as deep neural networks, to generalize across large state-action spaces. They have achieved excellent performances in a wide range of domains, including the games of Go and chess (Silver et al., 2017; Schrittwieser et al., 2020), robotics (Gu et al., 2017; Levine et al., 2018), and control systems (Degrave et al., 2022).

Several studies have studied the theoretical guarantees of tabular offline RL and proved near-optimal sample complexities in this setting (Xie et al., 2021b; Shi et al., 2022; Li et al., 2022). However, these algorithms cannot handle numerous real-world applications with large state spaces. Consequently, a significant body of research has shifted its focus to offline RL with function approximation. For example, several works have analyzed the sample efficiency of offline RL with linear function approximation under different MDP models, including linear MDPs and their variants (Jin et al., 2021b; Zanette et al., 2021; Min et al., 2021; Yin et al., 2022a). To handle nonlinear function class, a recent line of research considered offline RL with general function approximation (Chen and Jiang,

2019; Xie et al., 2021a; Zhan et al., 2022). While these algorithms have sample efficiency guarantees, they often require an inefficient planning phase or have a poor dependency on the function class complexity. For example, Xie et al. (2021a) proposed an information-theoretic algorithm that requires solving an optimization problem over all potential policy and corresponding version space, which includes all functions with lower Bellman error. To overcome this limitation, Xie et al. (2021a) proposed a practical implementation, as a cost, the algorithm have a poor dependency on the function class complexity. Recently, (Yin et al., 2022b) studied the general differentiable function class and propose a computation efficient algorithm (PFQL). However, their result also have an addition dependence on the dimension $d$ of the parameter.

Therefore, a natural question arises:

*Can we design a computationally efficient algorithm that achieves the minimax optimality with respect to the complexity of nonlinear function class?*

We give an affirmative answer to the above question in this work. Our contributions are listed as follows:

- We propose a pessimism-based algorithm PNLSVI designed for nonlinear function approximation, which strictly generalizes the existing pessimism-based algorithms for both linear and differentiable function approximation (Xiong et al., 2022; Yin et al., 2022b). Our algorithm is oracle-efficient, i.e., it is computationally efficient when there exists an efficient regression oracle and bonus oracle for the function class (e.g., generalized linear function class).

- We prove a data-dependent regret bound with the widely used $D^2$-divergence in online nonlinear RL regime, which is optimal with respect to the function class complexity. Our analysis closes the gap to optimality for differentiable function approximation, which was previously an open problem (Yin et al., 2022b).

- We introduce a novel uniform coverage assumption for general function approximation that is generalized over the assumption in Yin et al. (2022b). Our assumption bridges between the minimum-eigenvalue assumption used in linear models and the generalized dimension for nonlinear function class, offering new insights into the function approximation problem in RL.

**Notation:** In this work, we use lowercase letters to denote scalars and use lower and uppercase boldface letters to denote vectors and matrices respectively. For a vector $\mathbf{x} \in \mathbb{R}^d$ and matrix $\boldsymbol{\Sigma} \in \mathbb{R}^{d \times d}$, we denote by $\|\mathbf{x}\|_2$ the Euclidean norm and $\|\mathbf{x}\|_{\boldsymbol{\Sigma}} = \sqrt{\mathbf{x}^\top \boldsymbol{\Sigma} \mathbf{x}}$. For two sequences $\{a_n\}$ and $\{b_n\}$, we write $a_n = O(b_n)$ if there exists an absolute constant $C$ such that $a_n \leq Cb_n$, and we write $a_n = \Omega(b_n)$ if there exists an absolute constant $C$ such that $a_n \geq Cb_n$. We use $\widetilde{O}(\cdot)$ and $\widetilde{\Omega}(\cdot)$ to further hide the logarithmic factors. For any $a \leq b \in \mathbb{R}$, $x \in \mathbb{R}$, let $[x]_{[a,b]}$ denote the truncate function $a \cdot \mathbb{1}(x \leq a) + x \cdot \mathbb{1}(a \leq x \leq b) + b \cdot \mathbb{1}(b \leq x)$, where $\mathbb{1}(\cdot)$ is the indicator function. For a positive integer $n$, we use $[n] = \{1, 2, .., n\}$ to denote the set of integers from 1 to $n$.

## 2  Related Work

**RL with function approximation.**  As one of the simplest function approximation classes, linear representation in RL has been extensively studied in recent years (Jiang et al., 2017; Dann et al., 2018; Yang and Wang, 2019; Jin et al., 2020; Wang et al., 2020c; Du et al., 2019; Sun et al., 2019; Zanette et al., 2020a,b; Weisz et al., 2021; Yang and Wang, 2020; Modi et al., 2020; Ayoub et al., 2020; Zhou et al., 2021; He et al., 2021). Several assumptions on the linear structure of the underlying MDPs have been made in these works, ranging from the *linear MDP* assumption (Yang and Wang, 2019; Jin et al., 2020; Hu et al., 2022; He et al., 2022; Agarwal et al., 2022) to the *low Bellman-rank* assumption (Jiang et al., 2017) and the *low inherent Bellman error* assumption (Zanette et al., 2020b). Extending the previous theoretical guarantees to more general problem classes, RL with nonlinear function classes has garnered increased attention in recent years (Wang et al., 2020b; Jin et al., 2021a; Foster et al., 2021; Du et al., 2021; Agarwal and Zhang, 2022; Agarwal et al., 2022). Various complexity measures of function classes have been studied including Bellman rank (Jiang et al., 2017), Bellman-Eluder dimension (Jin et al., 2021a), Decision-Estimation Coefficient (Foster et al., 2021) and generalized Eluder dimension (Agarwal et al., 2022). Among these works, the setting in our paper is most related to Agarwal et al. (2022) where $D^2$-divergence (Gentile et al., 2022) was introduced in RL to indicate the uncertainty of a sample with respect to a particular sample batch.

**Offline tabular RL.** There is a line of works integrating the principle of pessimism to develop statistically efficient algorithms for offline tabular RL setting (Rashidinejad et al., 2021; Yin and Wang, 2021; Xie et al., 2021b; Shi et al., 2022; Li et al., 2022). More specifically, Xie et al. (2021b) utilized the variance of transition noise and proposed a nearly optimal algorithm based on pessimism and Bernstein-type bonus. Subsequently, Li et al. (2022) proposed a model-based approach that achieves minimax-optimal sample complexity without burn-in cost for tabular MDPs. Shi et al. (2022) also contributed by proposing the first nearly minimax-optimal model-free offline RL algorithm.

**Offline RL with linear function approximation.** Jin et al. (2021b) presented the initial theoretical results on offline linear MDPs. They introduced a pessimism-principled algorithmic framework for offline RL and proposed an algorithm based on LSVI (Jin et al., 2020). Min et al. (2021) subsequently considered offline policy evaluation (OPE) in linear MDPs, assuming independence between data samples across time steps to obtain tighter confidence sets and proposed an algorithm with optimal $d$ dependence. Yin et al. (2022a) took one step further and considered the policy optimization in linear MDPs, which implicitly requires the same independence assumption. Zanette et al. (2021) proposed an actor-critic-based algorithm that establishes pessimism principle by directly perturbing the parameter vectors in a linear function approximation framework. Recently, Xiong et al. (2022) proposed a novel uncertainty decomposition technique via a reference function, which leads to a minimax-optimal sample complexity bound for offline linear MDPs without additional assumptions.

**Offline RL with general function approximation.** Chen and Jiang (2019) critically examined the assumptions underlying value-function approximation methods and established an information-theoretic lower bound. Xie et al. (2021a) introduced the concept of Bellman-consistent pessimism, which enables sample-efficient guarantees by relying solely on the Bellman-completeness assumption. Uehara and Sun (2021) focused on model-based offline RL with function approximation under partial coverage, demonstrating that realizability in the function class and partial coverage are sufficient for policy learning. Zhan et al. (2022) proposed an algorithm that achieves polynomial sample complexity under the realizability and single-policy concentrability assumptions. Nguyen-Tang and Arora (2023) proposed a method of random perturbations and pessimism for neural function approximation. For differentiable function classes, Yin et al. (2022b) made advancements by improving the sample complexity with respect to the stage $H$. However, their result had an additional dependence on the dimension $d$ of the parameter space, whereas in linear function approximation, the dependence is typically on $\sqrt{d}$.

## 3 Preliminaries

In our work, we consider the inhomogeneous episodic Markov Decision Processes (MDP), which can be denoted by a tuple of $\mathcal{M}\big(\mathcal{S}, \mathcal{A}, H, \{r_h\}_{h=1}^{H}, \{\mathbb{P}_h\}_{h=1}^{H}\big)$. In specific, $\mathcal{S}$ is the state space, $\mathcal{A}$ is the finite action space, $H$ is the length of each episode. For each stage $h \in [H]$, $r_h : \mathcal{S} \times \mathcal{A} \to [0, 1]$ is the reward function[1] and $\mathbb{P}_h(s'|s, a)$ is the transition probability function, which denotes the probability for state $s$ to transfer to next state $s'$ with current action $a$. A policy $\pi := \{\pi_h\}_{h=1}^{H}$ is a collection of mappings $\pi_h$ from a state $s \in \mathcal{S}$ to the simplex of action space $\mathcal{A}$. For simplicity, we denote the state-action pair as $z := (s, a)$. For any policy $\pi$ and stage $h \in [H]$, we define the value function $V_h^\pi(s)$ and the action-value function $Q_h^\pi(s, a)$ as the expected cumulative rewards starting at stage $h$, which can be denoted as follows:

$$Q_h^\pi(s, a) = r_h(s, a) + \mathbb{E}\bigg[ \sum_{h'=h+1}^{H} r_{h'}\big(s_{h'}, \pi_{h'}(s_{h'})\big)\big|s_h = s, a_h = a \bigg], \; V_h^\pi(s) = Q_h^\pi\big(s, \pi_h(s)\big),$$

where $s_{h'+1} \sim \mathbb{P}_h(\cdot|s_{h'}, a_{h'})$ denotes the observed state at stage $h' + 1$. By this definition, the value function $V_h^\pi(s)$ and action-value function $Q_h^\pi(s, a)$ are bounded in $[0, H]$. In addition, we define the optimal value function $V_h^*$ and the optimal action-value function $Q_h^*$ as $V_h^*(s) = \max_\pi V_h^\pi(s)$ and $Q_h^*(s, a) = \max_\pi Q_h^\pi(s, a)$. We denote the corresponding optimal policy by $\pi^*$. For any function $V : \mathcal{S} \to \mathbb{R}$, we denote $[\mathbb{P}_h V](s, a) = \mathbb{E}_{s' \sim \mathbb{P}_h(\cdot|s,a)} V(s')$ and $[\text{Var}_h V](s, a) = [\mathbb{P}_h V^2](s, a) - \big([\mathbb{P}_h V](s, a)\big)^2$ for simplicity. For any function $f : \mathcal{S} \to \mathbb{R}$, we define the Bellman operator $\mathcal{T}_h$ as $\mathcal{T}_h f(s_h, a_h) = \mathbb{E}_{s_{h+1} \sim \mathbb{P}_h(\cdot|s_h, a_h)} [r_h(s_h, a_h) + f(s_{h+1})]$, where we use the shorthand $f(s) = \max_{a \in \mathcal{A}} f(s, a)$ for simplicity. Based on this definition, for every stage $h \in [H]$ and policy $\pi$, we

---

[1]While we study the deterministic reward functions for simplicity, it is not difficult to generalize our results to stochastic reward functions.

have the following Bellman equation for value functions $Q_h^\pi(s, a)$ and $V_h^\pi(s)$, as well as the Bellman optimality equation for optimal value functions:

$$Q_h^\pi(s_h, a_h) = \mathcal{T}_h V_{h+1}^\pi(s_h, a_h), \ Q_h^*(s_h, a_h) = \mathcal{T}_h V_{h+1}^*(s_h, a_h),$$

where $V_{H+1}^\pi(s) = V_{H+1}^*(s) = 0$. We also define the Bellman operator for second moment as $\mathcal{T}_{2,h} f(s_h, a_h) = \mathbb{E}_{s_{h+1} \sim \mathbb{P}_h(\cdot|s_h, a_h)} \left[ \left( r_h(s_h, a_h) + f(s_{h+1}) \right)^2 \right]$. For simplicity, we omit the subscripts $h$ in the Bellman operator without causing confusion.

**Offline Reinforcement Learning:** In offline RL, the agent only have access to a batch-dataset $D = \{s_h^k, a_h^k, r_h^k : h \in [H], k \in [K]\}$, which is collected by a behavior policy $\mu$, and the agent cannot interact with the environment. Given the batch dataset, the goal of offline RL is finding a near-optimal policy $\pi$ that minimize the sub-optimality $V_1^*(s) - V_1^\pi(s)$. In addition, for each stage $h$ and behavior policy $\mu$, we denote the induced distribution of the state-action pair as $d_h^\mu$.

**General Function Approximation:** In this work, we focus on a special class of episodic MDPs, where the value function satisfies the following completeness assumption.

**Assumption 3.1** ($\epsilon$-completeness under general function approximation, Agarwal et al. 2022). Given a general function class $\{\mathcal{F}_h\}_{h \in [H]}$, where each function class $\mathcal{F}_h$ is composed of functions $f_h : \mathcal{S} \times \mathcal{A} \to [0, L]$. We assume for each stage $h \in [H]$, and any function $V : \mathcal{S} \to [0, H]$, there exists functions $f_h, f_{2,h} \in \mathcal{F}_h$ such that

$$\max_{(s,a) \in \mathcal{S} \times \mathcal{A}} |f_h(s, a) - \mathcal{T}_h V(s, a)| \leq \epsilon, \text{ and } \max_{(s,a) \in \mathcal{S} \times \mathcal{A}} |f_{2,h}(s, a) - \mathcal{T}_{2,h} V(s, a)| \leq \epsilon.$$

In addition, for each stage $h \in [H]$, we assume there exist a function $f_h^* \in \mathcal{F}_h$ closed to the optimal value function such that $\|f_h^* - Q_h^*\|_\infty \leq \epsilon$. For simplicity, we assume $L = O(H)$ throughout the paper and denote $\mathcal{N} = \max_{h \in [H]} |\mathcal{F}_h|$.

To deal with general function class $\mathcal{F}$, Agarwal et al. (2022) introduce the following measure to capture the function class complexity for online learning.

**Definition 3.2** (Generalized Eluder dimension, Agarwal et al. 2022). Given $\lambda > 0$, a sequence of state-action pairs $Z = \{z_i\}_{i \in [K]}$ and a sequence of non-negative weights $\boldsymbol{\sigma} = \{\sigma_i\}_{i \in [K]}$. Let $\mathcal{F}$ be a function class consisting of functions $f : \mathcal{S} \times \mathcal{A} \to [0, L]$. The generalized Eluder dimension of $\mathcal{F}$ is given by $\dim_{\alpha, K}(\mathcal{F}) := \sup_{Z, \boldsymbol{\sigma}:|Z|=K, \boldsymbol{\sigma} \geq \alpha} \dim(\mathcal{F}, Z, \boldsymbol{\sigma})$, where

$$\dim(\mathcal{F}, Z, \boldsymbol{\sigma}) := \sum_{i=1}^K \min \left( 1, \frac{1}{\sigma_i^2} D_{\mathcal{F}}^2(z_i; z_{[i-1]}, \sigma_{[i-1]}) \right),$$

$$D_{\mathcal{F}}^2(z; z_{[k-1]}, \sigma_{[k-1]}) := \sup_{f_1, f_2 \in \mathcal{F}} \frac{(f_1(z) - f_2(z))^2}{\sum_{s \in [k-1]} \frac{1}{\sigma_s^2} (f_1(z_s) - f_2(z_s))^2 + \lambda}.$$

Here, the inequality $\boldsymbol{\sigma} \geq \alpha$ represents that $\sigma_i \geq \alpha$ holds for all $i \in [K]$ and we use the notation $z_{[i-1]}, \sigma_{[i-1]}$ to represent the sequences $\{z_s\}_{s=1}^{i-1}, \{\sigma_s\}_{s=1}^{i-1}$.

However, in offline RL, the proposed Generalized Eluder dimension fails to capture the relationship between function class $\mathcal{F}$ and the pre-collected dataset $\mathcal{D}$. To generalize this definition to offline environment, for a batch dataset $\mathcal{D} = \{(s_h^k, a_h^k, r_h^k)\}_{h,k=1}^{H,K}$ and a function class $\mathcal{F}_h$ consisting of functions $f : \mathcal{S} \times \mathcal{A} \to \mathbb{R}$. We denote $\mathcal{D}_h = \{(s_h^k, a_h^k, r_h^k)\}_{k \in [K]}$ as the subset of the dataset $D$ that corresponds to the observations collected up to stage $h$ in the MDP. Then for any weight function $\sigma_h(\cdot, \cdot) : \mathcal{S} \times \mathcal{A} \to \mathbb{R}$, we introduce the following $D^2$-divergence:

$$D_{\mathcal{F}_h}^2(z; \mathcal{D}_h; \sigma_h) = \sup_{f_1, f_2 \in \mathcal{F}_h} \frac{(f_1(z) - f_2(z))^2}{\sum_{k \in [K]} \frac{1}{(\sigma_h(z_h^k))^2} (f_1(z_h^k) - f_2(z_h^k))^2 + \lambda}.$$

**Data Coverage Assumption:** In offline RL, there exist a discrepancy between the state-action distribution generated by the behavior policy and the distribution from the learned policy. Under this situation, the distribution shift problem can cause the learned policy to perform poorly or even fail in offline RL. Therefore, we propose the following data coverage assumption to control the distribution shift.

**Algorithm 1** Pessimistic Nonlinear Least-Squares Value Iteration (PNLSVI)

---

**Require:** Input confidence parameters $\beta'_{1,h}, \beta'_{2,h}, \beta_h$ and $\epsilon > 0$.

1: **Initialize:** Split the input dataset into $\mathcal{D} = \{s_h^k, a_h^k, r_h^k\}_{k,h=1}^{K,H}, \mathcal{D}' = \{\bar{s}_h^k, \bar{a}_h^k, \bar{r}_h^k\}_{k,h=1}^{K,H}$ ; Set the value function $\widehat{f}_{H+1}(\cdot) = \widehat{f}'_{H+1}(\cdot) = 0$.

2: **for** stage $h = H, \ldots, 1$ **do**

3: $\quad \widetilde{f}'_h = \arg\min_{f_h \in \mathcal{F}_h} \sum_{k \in [K]} \left( f_h(\bar{s}_h^k, \bar{a}_h^k) - \bar{r}_h^k - \widehat{f}'_{h+1}(\bar{s}_{h+1}^k) \right)^2.$

4: $\quad \widetilde{g}'_h = \arg\min_{g_h \in \mathcal{F}_h} \sum_{k \in [K]} \left( g_h(\bar{s}_h^k, \bar{a}_h^k) - \left(\bar{r}_h^k + \widehat{f}'_{h+1}(\bar{s}_{h+1}^k)\right)^2 \right)^2.$

5: $\quad$ Use the bonus oracle (Definition 4.1) to calculate the bonus function
$\quad b'_h = \mathcal{B}(1, \mathcal{D}'_h, \mathcal{F}_h, \widetilde{f}'_h, \beta'_{1,h} + \beta'_{2,h}, \lambda, \epsilon),$

6: $\quad \widehat{f}'_h \leftarrow \{\widetilde{f}'_h - b'_h - \epsilon\}_{[0, H-h+1]};$

7: $\quad$ Construct the variance estimator
$\quad \widehat{\sigma}_h^2(s, a) = \max\left\{1, \widetilde{g}'_h(s, a) - (\widetilde{f}'_h(s, a))^2 - O\left(\frac{\sqrt{\log \mathcal{N}\mathcal{N}_b}H^3}{\sqrt{K\kappa}}\right)\right\}.$

8: **end for**

9: **for** stage $h = H, \ldots, 1$ **do**

10: $\quad \widetilde{f}_h = \arg\min_{f_h \in \mathcal{F}_h} \sum_{k \in [K]} \frac{1}{\widehat{\sigma}_h^2(s_h^k, a_h^k)} \left( f_h(s_h^k, a_h^k) - r_h^k - \widehat{f}_{h+1}(s_{h+1}^k) \right)^2$

11: $\quad$ Use the bonus oracle (Definition 4.1) to calculate the bonus function
$\quad b_h = \mathcal{B}(\widehat{\sigma}_h, \mathcal{D}_h, \mathcal{F}_h, \widetilde{f}_h, \beta_h, \lambda, \epsilon);$

12: $\quad \widehat{f}_h \leftarrow \{\widetilde{f}_h - b_h - \epsilon\}_{[0, H-h+1]};$

13: $\quad \widehat{\pi}_h(\cdot|s) = \arg\max_a \widehat{f}_h(s, a).$

14: **end for**

15: **Output:** $\widehat{\pi} = \{\widehat{\pi}_h\}_{h=1}^{H}.$

---

**Assumption 3.3** (Uniform Data Coverage). There exists a constant $\kappa > 0$, such that for any stage $h$ and functions $f_1, f_2 \in \mathcal{F}_h$, the following inequality holds,

$$\mathbb{E}_{\mu,h}\left[\left(f_1(s_h, a_h) - f_2(s_h, a_h)\right)^2\right] \geq \kappa \|f_1 - f_2\|_\infty^2,$$

where the state-action pair (at stage $h$) $(s_h, a_h)$ is stochastic generated from behavior policy $\mu$.

**Remark 3.4.** Data coverage assumption is widely used in offline RL to guarantee that the collected dataset contains enough information of the state-action space to learn an effective policy. In Yin et al. (2022b), they studied the general differentiable function, where the function class is defined as

$$\mathcal{F} := \left\{ f(\boldsymbol{\theta}, \boldsymbol{\phi}(\cdot, \cdot)) : \mathcal{S} \times \mathcal{A} \to \mathbb{R}, \boldsymbol{\theta} \in \Theta \right\}.$$

Under this definition, Yin et al. (2022b) introduce the following coverage assumption (Assumption 2.3) such that for all stage $h \in [H]$, there exists a constant $\kappa$,

$$\mathbb{E}_{\mu,h}\left[\left(f(\boldsymbol{\theta}_1, \boldsymbol{\phi}(s,a)) - f(\boldsymbol{\theta}_2, \boldsymbol{\phi}(s,a))\right)^2\right] \geq \kappa \|\boldsymbol{\theta}_1 - \boldsymbol{\theta}_2\|_2^2, \forall \boldsymbol{\theta}_1, \boldsymbol{\theta}_2 \in \Theta; \quad (*)$$

$$\mathbb{E}_{\mu,h}\left[\nabla f(\boldsymbol{\theta}, \boldsymbol{\phi}(s,a)) \nabla f(\boldsymbol{\theta}, \boldsymbol{\phi}(s,a))^\top\right] \succ \kappa I, \forall \boldsymbol{\theta} \in \Theta. \quad (**)$$

We can prove that our assumption is weaker than the first assumption (*). For the second assumption (**), there is no direct counterpart in the general setting.

In addition, for the linear function class, the coverage assumption in Yin et al. (2022b) will reduce to the following linear function coverage assumption (Wang et al., 2020a; Min et al., 2021; Yin et al., 2022a; Xiong et al., 2022).

$$\lambda_{\min}(\mathbb{E}_{\mu,h}[\phi(s,a)\phi(s,a)^\top]) = \kappa > 0, \forall h \in [H].$$

Therefore, our assumption is also weaker than the linear function coverage assumption when dealing with the linear function class. Due to space limitations, we provide the detailed proof in the appendix.

# 4 Algorithm

In this section, we provide a comprehensive and detailed description of our algorithm (PNLSVI), as displayed in Algorithm 1. In the sequel, we introduce the key ideas of the proposed algorithm.

## 4.1 Pessimistic Value Iteration Based Planning

Our algorithm operates in two distinct phases, Variance Estimate Phase and Pessimistic Planning Phase. At the beginning of the algorithm, the data-set is divided into two disjoint subsets $\mathcal{D}, \mathcal{D}'$, and each assigned to a specific phase.

The basic framework of our algorithm follows the pessimistic value iteration, which was initially introduced by Jin et al. (2021b). In details, for each stage $h \in [H]$, we construct the estimator value function $\widetilde{f}_h$ by solving the following variance-weighted ridge regression (Line 11):

$$\widetilde{f}_h = \operatorname*{argmin}_{f_h \in \mathcal{F}_h} \sum_{k \in [K]} \frac{1}{\widehat{\sigma}_h^2(s_h^k, a_h^k)} \left( f_h(s_h^k, a_h^k) - r_h^k - \widehat{f}_{h+1}(s_{h+1}^k) \right)^2,$$

where $\widehat{\sigma}_h^2$ is the estimated variance and will be discussed in section 4.2. In Line 12, we subtract the confidence bonus function $b_h$ from the estimator value function $\widetilde{f}_h$ to construct the pessimistic value function $\widehat{f}_h$. With the help of the confidence bonus function $b_h$, the pessimistic value function $\widehat{f}_h$ is almost a lower bound for the optimal value function $f_h^*$. The details of the bonus function and bonus oracle will be discussed in section 4.3.

Based on the pessimistic value function $\widehat{f}_h$ for stage $h$, we recursively perform the value iteration for the stage $h - 1$. Finally, we use the pessimistic value function $\widehat{f}_h$ to do planning and output the greedy policy with respect to the pessimistic value function $\widehat{f}_h$ (Line 13 - Line 15).

## 4.2 Variance Estimate Phase

In this phase, we provide a estimator for the variance $\widehat{\sigma}_h$ in the weighted ridge regression. According to the definition of Bellman operators $\mathcal{T}$ and $\mathcal{T}_2$, the variance of the function $\widehat{f}'_{h+1}$ for each state-action pair $(s, a)$ can be denoted by

$$[\mathrm{Var}_h \widehat{f}_{h+1}](s, a) = \mathcal{T}_{2,h} \widehat{f}'_{h+1}(s, a) - \left( \mathcal{T}_h \widehat{f}'_{h+1}(s, a) \right)^2.$$

Therefore, we need the evaluate the first-order and second-order moments for $\widehat{f}'_h$. We perform nonlinear least-squares regression separately for each of these moments. Specifically, in Line 3, we conduct regression to estimate the first-order moment.

$$\widetilde{f}'_h = \operatorname*{argmin}_{f_h \in \mathcal{F}_h} \sum_{k \in [K]} \left( f_h(\bar{s}_h^k, \bar{a}_h^k) - \bar{r}_h^k - \widehat{f}'_{h+1}(\bar{s}_{h+1}^k) \right)^2.$$

In Line 4, we perform regression for the second-order moment.

$$\widetilde{g}'_h = \operatorname*{argmin}_{g_h \in \mathcal{F}_h} \sum_{k \in [K]} \left( g_h(\bar{s}_h^k, \bar{a}_h^k) - \left( \bar{r}_h^k + \widehat{f}'_{h+1}(\bar{s}_{h+1}^k) \right)^2 \right)^2.$$

In this phase, we set the variance function to 1 for each state-action pair $(s, a)$ and derive an estimator with confidence radius $\beta'_{1,h}, \beta'_{2,h}$. Combing these two regression results and subtracting a confidence bonus function $b'_h$, we create a pessimistic estimator for the variance function (Lines 6 to 7).

## 4.3 Nonlinear Bonus Oracle

As we discussed in sections 4.1 and 4.2, we introduce a uncertainty bonus function to construct a pessimistic estimate of the value function. Unfortunately, for a general class, the uncertainty bonus may varies greatly across different state-action pair. Under this situation, the addition uncertainty bonus function will highly increase the complexity of the pessimistic function class, which make it difficult to construct a accurate estimation and may significant deteriorate the final performance. To address this issue, we assume there exist a function class $\mathcal{W}$ with cardinally $|\mathcal{W}| = \mathcal{N}_b$ and can approximate the bonus function well. In addition, we assume there exist a nonlinear bonus oracle (Agarwal and Zhang, 2022), which can output the approximate bonus function in the class $\mathcal{W}$ for each dataset $\mathcal{D}_h$.

**Definition 4.1** (Oracle for bonus function). For an offline dataset $\mathcal{D} = \{s_h^k, a_h^k, r_h^k\}_{h,k=1}^{H,K}$, given index $h \in [H]$, let $\mathcal{D}_h = \{(s_h^k, a_h^k, r_h^k)\}_{k \in [K]}$ denote the subset of the dataset $D$ that corresponds to the observations collected up to stage h in the MDP. $\widehat{\sigma}_h(\cdot, \cdot) : \mathcal{S} \times \mathcal{A} \to \mathbb{R}$ is a variance function. $\mathcal{F}_h$ is a function class such that $\widehat{f}_h \in \mathcal{F}_h$. The parameters $\beta_h, \lambda \geq 0$, error parameter $\epsilon \geq 0$. The bonus oracle $\mathcal{B}(\widehat{\sigma}, \mathcal{D}_h, \mathcal{F}_h, \widehat{f}_h, \beta_h, \lambda, \epsilon)$ outputs a bonus function $b_h(\cdot)$ such that

- $b_h : \mathcal{S} \times \mathcal{A} \to \mathbb{R}_{\geq 0}$ belongs to function class $\mathcal{W}$.

- $b_h(z_h) \geq \max \left\{ |f_h(z_h) - \widehat{f}_h(z_h)|, f_h \in \mathcal{F}_h : \sum_{k \in [K]} \frac{(f_h(z_h^k) - \widehat{f}_h(z_h^k))^2}{(\widehat{\sigma}_h(s_h^k, a_h^k))^2} \leq (\beta_h)^2 \right\}$ for any $z_h \in \mathcal{S} \times \mathcal{A}$.

- $b_h(z_h) \leq C \cdot \left( D_{\mathcal{F}_h}(z_h; \mathcal{D}_h; \widehat{\sigma}_h) \cdot \sqrt{(\beta_h)^2 + \lambda} + \epsilon \beta_h \right)$ for all $z_h \in \mathcal{S} \times \mathcal{A}$ with constant $0 < C \leq \infty$.

**Remark 4.2.** To address the concern of function class complexity, some previous studies (Xie et al., 2021a) have approached the problem differently. Instead of introducing a pointwise bonus in the estimated value function, they solve a complicated optimization problem to guarantee the optimism solely in the intial state. This method can prevent the complexity from bonus function, as a cost, they requires solving an optimization problem over all potential policy and corresponding version space, which includes all functions with lower Bellman error.

# 5  Main Results

In this section we prove an problem-dependent regret bound of Algorithm 1.

**Theorem 5.1.** Under Assumption 3.3, for $K \geq \widetilde{\Omega}\left( \frac{\log(\mathcal{N} \mathcal{N}_b) H^6}{\kappa^2} \right)$, if we set the parameters $\beta'_{1,h}, \beta'_{2,h} = \widetilde{O}(\sqrt{\log \mathcal{N} \mathcal{N}_b} H^2)$ and $\beta_h = \widetilde{O}(\sqrt{\log \mathcal{N}})$ in Algorithm 1, then with the probability of at least $1 - \delta$, for any state $s \in \mathcal{S}$, we have

$$V_1^*(s) - V_1^{\widehat{\pi}}(s) \leq \widetilde{O}(\sqrt{\log \mathcal{N}}) \sum_{h=1}^{H} \mathbb{E}_{\pi^*} \left[ D_{\mathcal{F}_h}(z_h; \mathcal{D}_h; [\mathbb{V}_h V_{h+1}^*](\cdot, \cdot)) | s_1 = s \right],$$

where $[\mathbb{V}_h V_{h+1}^*](s, a) = \max\{1, [\text{Var}_h V_{h+1}^*](s, a)\}$ is the truncated conditional variance.

**Remark 5.2.** When reduce to the linear MDP environment, the following function classes

$$\mathcal{F}_h^{\text{lin}} = \{ \langle \boldsymbol{\phi}_h(\cdot, \cdot), \boldsymbol{\theta}_h \rangle : \boldsymbol{\theta}_h \in \mathbb{R}^d, \|\boldsymbol{\theta}_h\|_2 \leq B_h \} \text{ for any } h \in [H],$$

satisfy the completeness assumption (Assumption 3.1) (Jin et al., 2020). Let $\mathcal{F}_h^{\text{lin}}(\epsilon)$ be a $\epsilon$-net of the linear function class $\mathcal{F}_h^{\text{lin}}$. In this case, the covering number satisfies $\log |\mathcal{F}_h^{\text{lin}}(\epsilon)| = \widetilde{O}(d)$ and the dependency of function class will reduce to $\widetilde{O}(\sqrt{\log \mathcal{N}}) = \widetilde{O}(\sqrt{d})$. For linear function class, Xiong et al. (2022) proposed the following regret guarantee,

$$V_1^*(s) - V_1^{\widehat{\pi}}(s) \leq \widetilde{O}(\sqrt{d}) \cdot \sum_{h=1}^{H} \mathbb{E}_{\pi^*} \left[ \|\boldsymbol{\phi}(s_h, a_h)\|_{\boldsymbol{\Sigma}_h^{*-1}} | s_1 = s \right],$$

where $\boldsymbol{\Sigma}_h^* = \sum_{k \in [K]} \boldsymbol{\phi}(s_h^k, a_h^k) \boldsymbol{\phi}(s_h^k, a_h^k)^\top / [\mathbb{V}_h V_{h+1}^*](s_h^k, a_h^k) + \lambda \mathbf{I}$. In comparison, we can prove the following inequality:

$$D_{\mathcal{F}_h^{\text{lin}}(\epsilon)}(z; \mathcal{D}_h; [\mathbb{V}_h V_{h+1}^*](\cdot, \cdot)) \leq \|\boldsymbol{\phi}_h(z)\|_{\boldsymbol{\Sigma}_h^{*-1}}.$$

This shows that Theorem 5.1 matches the optimal result in Xiong et al. (2022) for linear function class.

# 6  Key Techniques

In this section, we provide an overview of the key techniques in our algorithm design and analysis.

## 6.1  Variance Estimator with Nonlinear Function Class

The technique of variance-weighted ridge regression, first introduced in Zhou et al. (2021), has demonstrated its effectiveness in the online RL setting with linear function approximation. For offline

setting, Xiong et al. (2022) modified the variance-weighted ridge regression technique, and showed that using an accurate and independent variance estimator can improves the performance of the pessimistic value iteration (PEVI) algorithm (Jin et al., 2021b).

In our work, we extend this technique to general nonlinear function class $\mathcal{F}$, and use the following nonlinear least-squares regression to estimate the underlying value function:

$$\widetilde{f}_h = \underset{f_h \in \mathcal{F}_h}{\arg\min} \sum_{k \in [K]} \frac{1}{\widehat{\sigma}_h^2(s_h^k, a_h^k)} \left( f_h(s_h^k, a_h^k) - r_h^k - \widehat{f}_{h+1}(s_{h+1}^k) \right)^2.$$

For this regression, it is crucial to obtain a reliable evaluation for the variance of the estimated cumulative reward $r_h^k + \widehat{f}_{h+1}(s_{h+1}^k)$. According to the definition of Bellman operators $\mathcal{T}$ and $\mathcal{T}_2$, the variance of the function $\widehat{f}'_{h+1}$ for each state-action pair $(s, a)$ can be denoted by

$$[\mathrm{Var}_h \widehat{f}'_{h+1}](s, a) = \mathcal{T}_{2,h} \widehat{f}'_{h+1}(s, a) - \left( \mathcal{T}_h \widehat{f}'_{h+1}(s, a) \right)^2.$$

To evaluate the first and second moment for the Bellman operator, we perform nonlinear least-squares regression on a separate dataset $\mathcal{D}'$ with uniform weight ($\widehat{\sigma}_h(s, a) = 1$ for all state-action pair $(s, a)$).

For simplicity, we denote the empirical variance as $\mathbb{B}_h(s, a) = \widetilde{g}'_h(s, a) - \left( \widetilde{f}'_h(s, a) \right)^2$, and the difference between empirical variance $\mathbb{B}_h(s, a)$ with actually variance $[\mathrm{Var}_h \widehat{f}'_{h+1}](s, a)$ is upper bound by

$$\left| \mathbb{B}_h(s, a) - [\mathrm{Var}_h \widehat{f}'_{h+1}](s, a) \right| \leq \left| \widetilde{g}'_h(s, a) - \mathcal{T}_{2,h} \widehat{f}'_{h+1}(s, a) \right| + \left| \left( \widetilde{f}'_h(s, a) \right)^2 - \left( \mathcal{T}_h \widehat{f}'_{h+1}(s, a) \right)^2 \right|.$$

For these nonlinear function estimator, the following Lemmas provide coarse concentration properties for the first and second order Bellman operators.

**Lemma 6.1.** Given a stage $h \in [H]$, let $\widehat{f}'_{h+1}(\cdot, \cdot) \leq H$ be the estimated value function constructed in Algorithm 1 Line 6. By utilizing Assumption 3.1, there exists a function $\bar{f}'_h \in \mathcal{F}_h$, such that $|\bar{f}'_h(z_h) - \mathcal{T}_h \widehat{f}'_{h+1}(z_h)| \leq \epsilon$ holds for all state-action pair $z_h = (s_h, a_h)$. Then with the probability of at least $1 - \delta/4H$, it holds that $\sum_{k \in [K]} \left( \bar{f}'_h(\bar{z}_h^k) - \widetilde{f}'_h(\bar{z}_h^k) \right)^2 \leq (\beta'_{1,h})^2$, where $\beta'_{1,h} = \widetilde{O}\left( \sqrt{\log \mathcal{N} \mathcal{N}_b} H^2 \right)$, and $\widetilde{f}'_h$ is the estimated function for first-moment Bellman operator (Line 3 in Algorithm 1).

**Lemma 6.2.** Given a stage $h \in [H]$, let $\widehat{f}'_{h+1}(\cdot, \cdot) \leq H$ be the estimated value function constructed in Algorithm 1 Line 6. By utilizing Assumption 3.1, there exists a function $\bar{g}'_h \in \mathcal{F}_h$, such that $|\bar{g}'_h(z_h) - \mathcal{T}_{2,h} \widehat{f}'_{h+1}(z_h)| \leq \epsilon$ holds for all state-action pair $z_h = (s_h, a_h)$. Then with the probability of at least $1 - \delta/4H$, it holds that $\sum_{k \in [K]} \left( \bar{g}'_h(\bar{z}_h^k) - \widetilde{g}'_h(\bar{z}_h^k) \right)^2 \leq (\beta'_{2,h})^2$, where $\beta'_{2,h} = \widetilde{O}\left( \sqrt{\log \mathcal{N} \mathcal{N}_b} H^2 \right)$, and $\widetilde{g}'_h$ is the estimated function for second-moment Bellman operator (Line 4 in Algorithm 1).

Notice that all of the previous analysis focuses on the estimated function $\widehat{f}'_{h+1}$. By leveraging an induction procedure similar to existing works in the linear case (Jin et al., 2021b; Xiong et al., 2022), we can control the distance between the estimated function $\widehat{f}'_{h+1}$ and the optimal value function $f_h^*$. In details, with high probability, for all stage $h \in [H]$, the distance is upper bounded by $O\left( \sqrt{\log \mathcal{N} \mathcal{N}_b} H^3 / \sqrt{K \kappa} \right)$. This result allows us to further bound $[\mathrm{Var}_h \widehat{f}'_{h+1}](s, a)$ and $[\mathrm{Var}_h f_{h+1}^*](s, a)$.

Therefore, the concentration properties in Lemmas 6.1 and 6.2 enable us to construct the pessimistic variance estimator, which satisfies the following property:

$$[\mathbb{V}_h V_{h+1}^*](s, a) - \widetilde{O}\left( \frac{\sqrt{\log \mathcal{N} \mathcal{N}_b} H^3}{\sqrt{K \kappa}} \right) \leq \widehat{\sigma}_h^2(s, a) \leq [\mathbb{V}_h V_{h+1}^*](s, a). \tag{6.1}$$

where $[\mathbb{V}_h V_{h+1}^*](s, a) = \max\{1, [\mathrm{Var}_h V_{h+1}^*](s, a)\}$ is the truncated conditional variance. Compared with the results in the linear function class, we utilize the logarithm of the covering number of the function class as a substitute for the linear dimension $d$, which is a common technique in nonlinear function approximation.

## 6.2 Reference-Advantage Decomposition

The reference-advantage decomposition is a powerful technique to tackle the challenge of additional error from uniform concentration over whole function class $\mathcal{F}_h$. Such an analysis approach has been first studied in the online RL setting Azar et al. (2017); Zhang et al. (2021); Hu et al. (2022); He et al. (2022); Agarwal et al. (2022) and later in the offline environment by Xiong et al. (2022).

For offline RL, in the context of nonlinear function classes, without a explicit linear expression, the increased complexity of the function class structure poses a significant obstacle to effectively utilizing this technique. Previous works, such as Yin et al. (2022b), have struggled to adapt the reference-advantage decomposition to their nonlinear function class, resulting in a parameter space dependence that scales with $d$, instead of the optimal $\sqrt{d}$. We provide detailed insights into this approach as follows:

$$r_h(s,a) + \widehat{f}_{h+1}(s,a) - \mathcal{T}_h\widehat{f}_{h+1}(s,a) = \underbrace{r_h(s,a) + f_{h+1}^*(s,a) - \mathcal{T}_h f_{h+1}^*(s,a)}_{\text{Reference uncertainty}}$$

$$+ \underbrace{\widehat{f}_{h+1}(s,a) - f_{h+1}^*(s,a) - ([\mathbb{P}_h\widehat{f}_{h+1}](s,a) - [\mathbb{P}_h f_{h+1}^*](s,a))}_{\text{Advantage uncertainty}}.$$

We decompose the Bellman error into two parts: the Reference uncertainty and the Advantage uncertainty. For the first term, the optimal value function $f_{h+1}^*$ is fixed and not related to the pre-collected dataset, which circumvents additional uniform concentration over the whole function class and avoid any dependence on the function class size. For the second term, it is worth to notice that the distance between the estimated function $\widehat{f}_{h+1}'$ and the optimal value function $f_h^*$ is decreased as $O(1/\sqrt{K\kappa})$. Though, we still need to maintain the uniform convergence guarantee, the Advantage uncertainty is dominated by the Reference uncertainty when the number of episode $K$ is large enough. By integrating these results, we can prove a variance-weighted concentration inequality for Bellman operators.

**Lemma 6.3.** For each stage $h \in [H]$, assuming the variance estimator $\widehat{\sigma}_h$ satisfies (6.1), let $\widehat{f}_{h+1}(\cdot,\cdot) \le H$ be the estimated value function constructed in Algorithm 1 Line 12. By utilizing Assumption 3.1, there exist a function $\bar{f}_h \in \mathcal{F}_h$, such that $|\bar{f}_h(z_h) - \mathcal{T}_h\widehat{f}_{h+1}(z_h)| \le \epsilon$ holds for all state-action pair $z_h = (s_h, a_h)$. Then with the probability of at least $1 - \delta/4H$, it holds that $\sum_{k\in[K]} \frac{1}{(\widehat{\sigma}_h(z_h^k))^2} \left(\bar{f}_h(z_h^k) - \widetilde{f}_h(z_h^k)\right)^2 \le (\beta_h)^2$, where $\beta_h = \widetilde{O}(\sqrt{\log\mathcal{N}})$ and $\widetilde{f}_h$ is the estimated function from the weighted ridge regression (Line 10 in Algorithm 1).

After controlling the Bellman error, with a similar argument to Jin et al. (2021b); Xiong et al. (2022), we obtain the following lemma, which provide an upper bound for the regret.

**Lemma 6.4** (Regret Decomposition Property). If $|\mathcal{T}_h\widehat{f}_{h+1}(z) - \widetilde{f}_h(z)| \le b_h(z)$ holds for all stage $h \in [H]$ and state-action pair $z = (s,a) \in \mathcal{S} \times \mathcal{A}$, then the regret of Algorithm 1 can be bounded as

$$V_1^*(s) - V_1^{\widehat{\pi}}(s) \le 2\sum_{h=1}^{H}\mathbb{E}_{\pi^*}\left[b_h(s_h, a_h) \mid s_1 = s\right].$$

Here, the expectation $\mathbb{E}_{\pi^*}$ is with respect to the trajectory induced by $\pi^*$ in the underlying MDP.

Combing the results in Lemmas 6.3 and 6.4, we have proved Theorem 5.1.

## 7 Conclusion and Future Work

In this paper, we present PNLSVI, an oracle-efficient algorithm for offline RL with non-linear function approximation. It achieves minimax optimal problem-dependent regret when specialized to linear function approximation.

Regarding future work, we observe that instead of using the uniform coverage assumption, a series of works, such as (Liu et al., 2020; Xie et al., 2021a; Uehara and Sun, 2021; Zhan et al., 2022), only relies on partial coverage assumption. In these works, the offline data distribution only encompasses the state-action distribution of a select high-quality comparator policy $\pi^*$. It would be of significant interest to investigate whether it's possible to design practical algorithms for nonlinear function classes under this weaker partial coverage assumption, while still preserving the inherent efficiency found in linear function approximation.

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
