can be denoted by

$$\mathcal{F} := \Big\{ f\big(\boldsymbol{\theta}, \boldsymbol{\phi}(\cdot, \cdot)\big) : \mathcal{X} \times \mathcal{A} \to \mathbb{R}, \boldsymbol{\theta} \in \Theta \Big\}.$$

 In this definition, $\Psi$ is a compact subset and $\boldsymbol{\phi}(\cdot, \cdot) : \mathcal{X} \times \mathcal{A} \to \Psi \subseteq \mathbb{R}^m$ is a feature map. The
 parameter space $\Theta$ is a compact subset $\Theta \subseteq \mathbb{R}^d$. The function $f : \mathbb{R}^d \times \mathbb{R}^m \to \mathbb{R}$ satisfies the
 following smoothness conditions:

   - For any vector $\boldsymbol{\phi} \in \mathbb{R}^m$, $f(\boldsymbol{\theta}, \boldsymbol{\phi})$ is third-time differentiable with respect to the parameter $\boldsymbol{\theta}$.
   - Functions $f, \partial_{\boldsymbol{\theta}} f, \partial^2_{\boldsymbol{\theta}, \boldsymbol{\theta}} f, \partial^3_{\boldsymbol{\theta}, \boldsymbol{\theta}, \boldsymbol{\theta}} f$ are jointly continuous for $(\boldsymbol{\theta}, \boldsymbol{\phi})$.

 Under this definition, Yin et al. (2022b) introduce the following coverage assumption (Assumption
 2.3) such that for all stage $h \in [H]$, there exists a constant $\kappa$,

$$\mathbb{E}_{\mu,h}\left[ \big(f\left(\boldsymbol{\theta}_1, \boldsymbol{\phi}(x, a)\right) - f(\boldsymbol{\theta}_2, \boldsymbol{\phi}(x, a))\big)^2 \right] \geq \kappa \|\boldsymbol{\theta}_1 - \boldsymbol{\theta}_2\|_2^2, \forall \boldsymbol{\theta}_1, \boldsymbol{\theta}_2 \in \Theta; (*)$$

$$\mathbb{E}_{\mu,h}\left[ \nabla f(\boldsymbol{\theta}, \boldsymbol{\phi}(x, a)) \nabla f(\boldsymbol{\theta}, \boldsymbol{\phi}(x, a))^\top \right] \succ \kappa I, \forall \boldsymbol{\theta} \in \Theta. (**)$$

 It is worth noting that our assumption 3.3 is weaker than this assumption. For any compact sets $\Theta, \Psi$
 and continuous function $f$, there always exist a constant $\kappa_0 > 0$ such that $f$ is $\kappa_0$-Lipschitz with
 respect to the parameter $\boldsymbol{\theta}$,i.e:

$$|f(\boldsymbol{\theta}_1, \boldsymbol{\phi}) - f(\boldsymbol{\theta}_2, \boldsymbol{\phi})| \leq \kappa_0 \|\boldsymbol{\theta}_1 - \boldsymbol{\theta}_2\|_2, \forall \boldsymbol{\theta}_1, \boldsymbol{\theta}_2 \in \Theta, \boldsymbol{\phi} \in \Psi.$$

 Therefore, the coverage assumption in Yin et al. (2022b) implies that

$$\mathbb{E}_{\mu,h}\left[ \big(f(\boldsymbol{\theta}_1, \boldsymbol{\phi}(\cdot, \cdot)) - f(\boldsymbol{\theta}_2, \boldsymbol{\phi}(\cdot, \cdot))\big)^2 \right] \geq \kappa \|\boldsymbol{\theta}_1 - \boldsymbol{\theta}_2\|_2^2$$
$$\geq \frac{\kappa}{\kappa_0^2} \sup_{(x, a) \in \mathcal{X} \times \mathcal{A}} \big(f(\boldsymbol{\theta}_1, \boldsymbol{\phi}(x, a)) - f(\boldsymbol{\theta}_2, \boldsymbol{\phi}(x, a))\big)^2.$$

 Thus, our assumption is weaker than the first assumption (*). For the second assumption (**), there
 is no direct counterpart in the general setting.

 In addition, for the linear function class, the coverage assumption in Yin et al. (2022b) will reduce to
 the following linear function coverage assumption(Wang et al., 2020a; Min et al., 2021; Yin et al.,
 2022a; Xiong et al., 2022).

$$\lambda_{\min}(\mathbb{E}_{\mu,h}[\phi(x, a)\phi(x, a)^\top]) = \kappa > 0, \ \forall h \in [H].$$

 Therefore, our assumption is also weaker than the linear function coverage assumption when dealing
 with the linear function class.

 # B  Proof of Theorem 5.1

 We need the following lemmas to prove Theorem 5.1. To start with, we prove the result that our data
 coverage assumption (Assumption 3.3) can lead to an upper bound of the $D^2$-divergence for large
 dataset.

 **Lemma B.1.** Let $\mathcal{D}_h$ be the dataset satisfying Assumption 3.3. When the size of data set satisfies
 $K \geq \widetilde{\Omega}\left(\frac{\log \mathcal{N}}{\kappa^2}\right)$, with probability at least $1 - \delta$, for each state-action pair $z$, we have

$$D_{\mathcal{F}_h}(z, \mathcal{D}_h, 1) = \widetilde{O}\left(\frac{1}{\sqrt{K\kappa}}\right).$$

 **Lemma B.2.** Let $\mathcal{D}_h$ be a dataset satisfying Assumption 3.3. When the size of data set satisfies
 $K \geq \widetilde{\Omega}\left(\frac{\log \mathcal{N}}{\kappa^2}\right)$, $\widehat{\sigma}_h \leq H$, with probability at least $1 - \delta$, for each state-action pair $z$, we have

$$D_{\mathcal{F}_h}(z, \mathcal{D}_h, \widehat{\sigma}_h) = \widetilde{O}\left(\frac{H}{\sqrt{K\kappa}}\right).$$

The following lemmas show the confidence radius for the first and second-order Bellman error.

**Lemma B.3** (Restatement of Lemma 6.1). At stage $h \in [H]$, the estimated value function $\widehat{f}'_{h+1}$ in Algorithm 1 is bounded by $H$. According to Assumption 3.1, there exists some function $\bar{f}'_h \in \mathcal{F}_h$, such that $|\bar{f}'_h(z_h) - \mathcal{T}_h \widehat{f}'_{h+1}(z_h)| \leq \epsilon$ for all $z_h = (s_h, a_h)$. Then with probability at least $1 - \delta/(4H^2)$, the following inequality holds:

$$\sum_{k \in [K]} \left( \bar{f}'_h(\bar{z}^k_h) - \widetilde{f}'_h(\bar{z}^k_h) \right)^2 \leq (\beta'_{1,h})^2,$$

where $\beta'_{1,h} = \widetilde{O}(\sqrt{\log \mathcal{N} \mathcal{N}_b} H)$.

The following lemma for second-order function approximation parallels the lemma we have proved.

**Lemma B.4** (Restatement of Lemma 6.2). At stage $h \in [H]$, the estimated value function $\widehat{f}'_{h+1}$ in Algorithm 1 is bounded by $H$. According to Assumption 3.1, there exists some functions $\bar{g}'_h \in \mathcal{F}_h$, such that $|\bar{g}_h(z_h) - \mathcal{T}_{2,h} \widehat{f}_{h+1}(z_h)| \leq \epsilon$ for all $z_h = (x_h, a_h)$. Then with probability at least $1 - \delta/(4H^2)$, the following inequality holds:

$$\sum_{k \in [K]} \left( \bar{g}'_h(\bar{z}^k_h) - \widetilde{g}'_h(\bar{z}^k_h) \right)^2 \leq (\beta'_{2,h})^2,$$

where $\beta'_{2,h} = \widetilde{O}(\sqrt{\log \mathcal{N} \mathcal{N}_b} H^2)$.

Using Lemma B.3 and Lemma B.4, we can prove a high probability bound of the variance estimator. We first recall the definition of the variance estimator.

$$\widetilde{f}'_h = \operatorname*{argmin}_{f_h \in \mathcal{F}_h} \sum_{k \in [K]} \left( f_h(\bar{s}^k_h, \bar{a}^k_h) - \bar{r}^k_h - \widehat{f}'_{h+1}(\bar{s}^k_{h+1}) \right)^2$$

$$\widetilde{g}'_h = \operatorname*{argmin}_{g_h \in \mathcal{F}_h} \sum_{k \in [K]} \left( g_h(\bar{s}^k_h, \bar{a}^k_h) - \left( \bar{r}^k_h + \widehat{f}'_{h+1}(\bar{s}^k_{h+1}) \right)^2 \right)^2 .$$

We then employ the following variance estimator:

$$\widehat{\sigma}^2_h(s,a) := \max \left\{ 1, \widetilde{g}'_h(s,a) - \left( \widetilde{f}'_h(s,a) \right)^2 - \widetilde{O}\left( \frac{\sqrt{\log \mathcal{N} \mathcal{N}_b} H^3}{\sqrt{K\kappa}} \right) \right\}.$$

The following lemma shows our constructed estimator is closed to the actual variance of the optimal value function $[\mathbb{V}_h V^*_{h+1}](s,a)$.

**Lemma B.5.** with probability at least $1 - \delta/2$, for any $h \in [H]$, the variance estimator designed above satisfies:

$$[\mathbb{V}_h V^*_{h+1}](s,a) - \widetilde{O}\left( \frac{\sqrt{\log \mathcal{N} \mathcal{N}_b} H^3}{\sqrt{K\kappa}} \right) \leq \widehat{\sigma}^2_h(s,a) \leq [\mathbb{V}_h V^*_{h+1}](s,a).$$

With the property of $\widehat{\sigma}_h$ in Lemma B.5, we can prove a variance weighted version of concentration inequality.

**Lemma B.6** (Restatement of Lemma 6.3). Suppose the variance function $\widehat{\sigma}_h$ satisfies the inequality in Lemma B.5. at stage $h \in [H]$, the estimated value function $\widehat{f}_{h+1}$ in Algorithm 1 is bounded by $H$. According to Assumption 3.1, there exists some function $\bar{f}_h \in \mathcal{F}_h$, such that $|\bar{f}_h(z_h) - \mathcal{T}_h \widehat{f}_{h+1}(z)| \leq \epsilon$ for all $z_h = (s_h, a_h)$. Then with probability at least $1 - \delta/2$, the following inequality holds for all stage $h \in [H]$ simultaneously,

$$\sum_{k \in [K]} \frac{1}{(\widehat{\sigma}_h(z^k_h))^2} \left( \bar{f}_h(z^k_h) - \widetilde{f}_h(z^k_h) \right)^2 \leq (\beta_h)^2.$$

Finally, we start the proof of Theorem 5.1.

*Proof of Theorem 5.1.* For any state-action pair $z = (s, a) \in \mathcal{S} \times \mathcal{A}$, we have

$$\left| \mathcal{T}_h \widehat{f}_{h+1}(z) - \widetilde{f}_h(z) \right| \leq \left| \mathcal{T}_h \widehat{f}_{h+1}(z) - \bar{f}_h(z) \right| + \left| \bar{f}_h(z) - \widetilde{f}_h(z) \right|$$
$$\leq \epsilon + b_h(z),$$

where we bound the first term with the Bellman completeness assumption (Assumption 3.1). For the second term, we use the bonus oracle (Definition 4.1) and Lemma B.6. Therefore, using Lemma D.2 we have

$$V_1^*(s) - \widehat{V}_1(s) \leq 2 \sum_{h=1}^H \mathbb{E}_{\pi^*} \left[ b_h(s_h, a_h) \mid s_1 = s \right] + 2\epsilon H$$

$$\leq \sum_{h=1}^H \mathbb{E}_{\pi^*} \left[ D_{\mathcal{F}_h}(z_h; \mathcal{D}_h; \widehat{\sigma}_h) \cdot \sqrt{(\beta_h)^2 + \lambda} \mid s_1 = s \right] + 2\epsilon H$$

$$\leq \widetilde{O}\left( \sqrt{\log \mathcal{N}} \right) \sum_{h=1}^H \mathbb{E}_{\pi^*} \left[ D_{\mathcal{F}_h}(z_h; \mathcal{D}_h; \widehat{\sigma}_h) \mid s_1 = s \right]$$

$$\leq \widetilde{O}\left( \sqrt{\log \mathcal{N}} \right) \sum_{h=1}^H \mathbb{E}_{\pi^*} \left[ D_{\mathcal{F}_h}\left( z_h; \mathcal{D}_h; [\mathbb{V}_h V_{h+1}^*](\cdot, \cdot) \right) \mid s_1 = s \right].$$

Here the second inequality holds because of our choice of bonus function (Definition 4.1). We use the definition of $\beta_h = \widetilde{O}\left( \sqrt{\log \mathcal{N}} \right)$ in the third inequality. Finally, due to Lemma B.5, with probability at least $1 - \delta$, for any $z \in \mathcal{S} \times \mathcal{A}$, we have $\widehat{\sigma}_h(z) \leq [\mathbb{V}_h V_{h+1}^*](z)$. Therefore, using the fact that $D^2$-divergence is increasing with respect to the variance function (Definition 3.2), we have proved the last inequality. $\square$

# C  Proof of the Lemmas in Section B

## C.1  Proof of Lemma B.1 and Lemma B.2

*Proof of Lemma B.1.* From the definition of $D^2$ divergence (Definition 3.2), we have

$$D_{\mathcal{F}_h}^2(z; \mathcal{D}_h; 1) = \sup_{f_1, f_2 \in \mathcal{F}_h} \frac{(f_1(z) - f_2(z))^2}{\sum_{k \in [K]} \left( f_1(z_h^k) - f_2(z_h^k) \right)^2 + \lambda} \tag{C.1}$$

By Hoeffding inequality (Lemma D.3), with probability at least $1 - \delta/(\mathcal{N}^2)$, we have

$$\sum_{k \in [K]} \left( f_1(z_h^k) - f_2(z_h^k) \right)^2 - K \mathbb{E}_{\mu, h} \left[ (f_1(z_h) - f_2(z_h))^2 \right] \geq -2\sqrt{2K \log(\mathcal{N}^2/\delta)} \cdot \|f_1 - f_2\|_\infty^2.$$

Hence, after taking a union bound, we have with probability at least $1 - \delta$, for all $f_1, f_2 \in \mathcal{F}_h$,

$$\sum_{k \in [K]} \left( f_1(z_h^k) - f_2(z_h^k) \right)^2 \geq K \mathbb{E}_{\mu, h} \left[ (f_1(z_h) - f_2(z_h))^2 \right] - 2\sqrt{2K \log(\mathcal{N}^2/\delta)} \cdot \|f_1 - f_2\|_\infty^2$$

$$\geq K \cdot \kappa \|f_1 - f_2\|_\infty^2 - 2\sqrt{2K \log(\mathcal{N}^2/\delta)} \cdot \|f_1 - f_2\|_\infty^2, \tag{C.2}$$

where the second inequality holds due to Assumption 3.3. Substituting (C.2) into (C.1), when the size of dataset $K \geq \widetilde{\Omega}\left( \frac{\log \mathcal{N}}{\kappa^2} \right)$, we have

$$D_{\mathcal{F}_h}^2(z; \mathcal{D}_h; 1) \leq \sup_{f_1, f_2 \in \mathcal{F}_h} \frac{(f_1(z) - f_2(z))^2}{\frac{1}{2} K \cdot \kappa \|f_1 - f_2\|_\infty^2 + \lambda} = \widetilde{O}\left( \frac{1}{K\kappa} \right).$$

$\square$

545 *Proof of Lemma B.2.* From the definition of $D^2$ divergence, we have

$$D^2_{\mathcal{F}_h}(z; \mathcal{D}_h; \widehat{\sigma}_h) = \sup_{f_1, f_2 \in \mathcal{F}_h} \frac{(f_1(z) - f_2(z))^2}{\sum_{k \in [K]} \frac{1}{(\widehat{\sigma}_h(z_h^k))^2} \left(f_1(z_h^k) - f_2(z_h^k)\right)^2 + \lambda} \tag{C.3}$$

546 By Hoeffding inequality (Lemma D.3), with probability at least $1 - \delta/(\mathcal{N}^2)$,

$$\sum_{k \in [K]} \left(f_1(z_h^k) - f_2(z_h^k)\right)^2 - K \mathbb{E}_{\mu,h} \left[(f_1(z_h) - f_2(z_h))^2\right] \geq -2\sqrt{2K \log(\mathcal{N}^2/\delta)} \cdot \|f_1 - f_2\|_\infty^2.$$

547 Hence, after taking a union bound, we have with probability at least $1 - \delta$, for all $f_1, f_2 \in \mathcal{F}_h$,

$$\sum_{k \in [K]} \frac{1}{(\widehat{\sigma}_h(z_h^k))^2} \left(f_1(z_h^k) - f_2(z_h^k)\right)^2$$

$$\geq \frac{1}{H^2} \left(K \mathbb{E}_{\mu,h} \left[(f_1(z_h) - f_2(z_h))^2\right] - 2\sqrt{2K \log(\mathcal{N}^2/\delta)} \cdot \|f_1 - f_2\|_\infty^2\right)$$

$$\geq \frac{1}{H^2} \left(K \cdot \kappa \|f_1 - f_2\|_\infty^2 - 2\sqrt{2K \log(\mathcal{N}^2/\delta)} \cdot \|f_1 - f_2\|_\infty^2\right), \tag{C.4}$$

548 where we use Assumption 3.3 and substituting (C.4) into (C.3), when $K \geq \widetilde{\Omega}\left(\frac{\log \mathcal{N}}{\kappa}\right)$, we have

$$D^2_{\mathcal{F}_h}(z; \mathcal{D}_h; \widehat{\sigma}_h) \leq \sup_{f_1, f_2 \in \mathcal{F}_h} \frac{H^2(f_1(z) - f_2(z))^2}{\frac{1}{2} K \cdot \kappa \|f_1 - f_2\|_\infty^2 + \lambda} = \widetilde{O}\left(\frac{H^2}{K\kappa^2}\right).$$

549 $\qquad\qquad\qquad\qquad\qquad\qquad\qquad\qquad\qquad\qquad\qquad\qquad\qquad\qquad\qquad\qquad\qquad\qquad\qquad\qquad\square$

## C.2 Proof of Lemma B.3

551 We need to prove the following concentration inequality first.

552 **Lemma C.1.** Based on the dataset $\mathcal{D}' = \{\bar{s}_h^k, \bar{a}_h^k, \bar{r}_h^k\}_{k,h=1}^{K,H}$, we define the filtration

$$\bar{\mathcal{H}}_h^k = \sigma\left(\bar{s}_1^1, \bar{a}_1^1, \bar{r}_1^1, \bar{s}_2^1, \ldots, \bar{r}_H^1, \bar{s}_{H+1}^1; \bar{s}_1^2, \bar{a}_1^2, \bar{r}_1^2, \bar{s}_2^2, \ldots, \bar{r}_H^2, \bar{s}_{H+1}^2; \ldots; \bar{s}_1^k, \bar{a}_1^k, \bar{r}_1^k, \bar{s}_2^k, \ldots, \bar{r}_h^k, \bar{s}_{h+1}^k\right).$$

553 For any fixed functions $f, f' : \mathcal{S} \to [0, L]$, we make the following definitions:

$$\bar{\eta}_h^k[f'] := f'(\bar{s}_{h+1}^k) - [\mathbb{P}_h f'](\bar{s}_h^k, \bar{a}_h^k)$$

$$\bar{D}_h^k[f, f'] := 2\bar{\eta}_h^k[f'] \left(f(\bar{z}_h^k) - \mathcal{T}_h f'(\bar{z}_h^k)\right).$$

554 Then with probability at least $1 - \delta/(4H^2\mathcal{N}^2\mathcal{N}_b^2)$, the following inequality holds,

$$\sum_{k \in [K]} \bar{D}_h^k[f, f'] \leq (24L + 5)i^2(\delta) + \frac{\sum_{k \in [K]} \left(f(\bar{z}_h^k) - \mathcal{T}_h f'(\bar{z}_h^k)\right)^2}{2},$$

555 where $i(\delta) = \sqrt{2 \log \frac{\mathcal{N}\mathcal{N}_b H(2\log(4LK)+2)(\log(2L)+2)}{\delta}}$.

556 *Proof.* We use Lemma D.1, with the following conditions:

$\bar{D}_h^k[f, f']$ is adapted to the filtration $\bar{\mathcal{H}}_h^k$ and $\mathbb{E}\left[\bar{D}_h^k[f, f'] \mid \bar{\mathcal{H}}_h^{k-1}\right] = 0$.

$\left|\bar{D}_h^k[f, f']\right| \leq 2\left|\bar{\eta}_h^k\right| \max_z |f(z) - \mathcal{T}_h f'(z)| \leq 4L^2 = M.$

$$\sum_{k \in [K]} \mathbb{E}\left[\left(\bar{D}_h^k[f, f']\right)^2 \Big| \bar{z}_h^k\right] = \sum_{k \in [K]} \mathbb{E}\left[4\left(\bar{\eta}_h^k[f']\right)^2 \Big| \bar{z}_h^k\right] \left(f(\bar{z}_h^k) - \mathcal{T}_h f'(\bar{z}_h^k)\right)^2 \leq (4LK)^2 = V^2.$$

557 On the other hand,

$$\sum_{k \in [K]} \mathbb{E}\left[\left(\bar{D}_h^k[f, f']\right)^2 \Big| \bar{z}_h^k\right] = \sum_{k \in [K]} \mathbb{E}\left[4\left(\bar{\eta}_h^k[f']\right)^2 \Big| \bar{z}_h^k\right] \left(f(\bar{z}_h^k) - \mathcal{T}_h f'(\bar{z}_h^k)\right)^2$$

$$\leq 8L^2 \sum_{k \in [K]} \left(f(\bar{z}_h^k) - \mathcal{T}_h f'(\bar{z}_h^k)\right)^2.$$

Then using Lemma D.1 with $v = 1$, $m = 1$, with high probability, we have:

$$\sum_{k \in [K]} 2\bar{\eta}_h^k[f'] \left( f(\bar{z}_h^k) - \mathcal{T}_h f'(\bar{z}_h^k) \right) \leq i(\delta) \sqrt{2(2 \cdot 8L^2) \sum_{k \in [K]} \left( f(\bar{z}_h^k) - \mathcal{T}_h f'(\bar{z}_h^k) \right)^2}$$
$$+ \frac{2}{3} i^2(\delta) + \frac{4}{3} i^2(\delta) \cdot 4L^2$$
$$\leq (24L^2 + 5)i^2(\delta) + \frac{\sum_{k \in [K]} \left( f(\bar{z}_h^k) - \mathcal{T}_h f'(\bar{z}_h^k) \right)^2}{2}.$$

$\square$

*Proof of Lemma B.3.* Let $(\beta'_{1,h})^2 = (24L^2 + 5)i^2(\delta) + 8KL\epsilon$. We define the event $\mathcal{E}'_{1,h} :=$ $\left\{ \sum_{k \in [K]} \left( \bar{f}'_h(\bar{z}_h^k) - \widetilde{f}'_h(\bar{z}_h^k) \right)^2 > (\beta'_{1,h})^2 \right\}$. The following inequality will be useful in our proof.

$$\sum_{k \in [K]} \left( \bar{f}'_h(\bar{z}_h^k) - \widetilde{f}'_h(\bar{z}_h^k) \right)^2 = \sum_{k \in [K]} \left[ \left( \bar{r}_h^k + \widehat{f'_{h+1}}(\bar{s}_{h+1}^k) - \bar{f}'_h(\bar{z}_h^k) \right) + \left( \widetilde{f}'_h(\bar{z}_h^k) - \bar{r}_h^k - \widehat{f'_{h+1}}(\bar{s}_{h+1}^k) \right) \right]^2$$

$$= \sum_{k \in [K]} \left( \bar{r}_h^k + \widehat{f'_{h+1}}(\bar{s}_{h+1}^k) - \bar{f}'_h(\bar{z}_h^k) \right)^2 + \sum_{k \in [K]} \left( \widetilde{f}'_h(\bar{z}_h^k) - \bar{r}_h^k - \widehat{f'_{h+1}}(\bar{s}_{h+1}^k) \right)^2$$

$$+ 2 \sum_{k \in [K]} \left( \bar{r}_h^k + \widehat{f'_{h+1}}(\bar{s}_{h+1}^k) - \bar{f}'_h(\bar{z}_h^k) \right) \left( \widetilde{f}'_h(\bar{z}_h^k) - \bar{r}_h^k - \widehat{f'_{h+1}}(\bar{s}_{h+1}^k) \right)$$

$$\leq 2 \sum_{k \in [K]} \left( \bar{r}_h^k + \widehat{f'_{h+1}}(\bar{s}_{h+1}^k) - \bar{f}'_h(\bar{z}_h^k) \right)^2$$

$$+ 2 \sum_{k \in [K]} \left( \bar{r}_h^k + \widehat{f'_{h+1}}(\bar{s}_{h+1}^k) - \bar{f}'_h(\bar{z}_h^k) \right) \left( \widetilde{f}'_h(\bar{z}_h^k) - \bar{r}_h^k - \widehat{f'_{h+1}}(\bar{s}_{h+1}^k) \right)$$

$$\leq 2 \sum_{k \in [K]} \left( \bar{r}_h^k + \widehat{f'_{h+1}}(\bar{s}_{h+1}^k) - \bar{f}'_h(\bar{z}_h^k) \right) \left( \widetilde{f}'_h(\bar{z}_h^k) - \bar{f}'_h(\bar{z}_h^k) \right).$$

Here we use our choice of $\widetilde{f}'_h$, i.e. $\widetilde{f}'_h = \text{argmin}_{f_h \in \mathcal{F}_h} \sum_{k \in [K]} \left( f_h(\bar{s}_h^k, \bar{a}_h^k) - \bar{r}_h^k - \widehat{f'_{h+1}}(\bar{s}_{h+1}^k) \right)^2$.
Next, we will use Lemma C.1. For any fixed $h$, let $f = \widetilde{f}'_h \in \mathcal{F}_h$, $f' = \widehat{f'_{h+1}} = \{\widetilde{f} - \epsilon\}_{[0, H-h+1]}$,
where $\widetilde{f} = \widetilde{f}'_h - b'_h \in \mathcal{F}_h - \mathcal{W}$. Following the construction in Lemma C.1, we define

$$\bar{\eta}_h^k[f'] = \bar{r}_h^s + f'(\bar{s}_{h+1}^k) - \mathbb{E} \left[ \bar{r}_h^k + f'(\bar{s}_{h+1}^k) | \bar{z}_h^k \right],$$
$$\text{and } \bar{D}_h^k[f, f'] = 2\bar{\eta}_h^k[f'] \left( f(\bar{z}_h^k) - \mathcal{T}_h \bar{f}'(\bar{z}_h^k) \right).$$

Due to the result of Lemma C.1, taking a union bound, we have with probability at least $1 - \delta/(4H^2)$, the following inequality holds,

$$\sum_{k \in [K]} \bar{D}_h^k[f, f'] \leq (24L^2 + 5)i^2(\delta) + \frac{\sum_{k \in [K]} \left( f(\bar{z}_h^k) - \mathcal{T}_h f'(\bar{z}_h^k) \right)^2}{2}. \tag{C.5}$$

567   Therefore, with probability at least $1 - \delta/(4H^2)$, we have

$$2 \sum_{k \in [K]} \left( \bar{r}_h^k + \widehat{f}_{h+1}'(\bar{s}_{h+1}^k) - \bar{f}_h'(\bar{z}_h^k) \right) \left( \widetilde{f}_h'(\bar{z}_h^k) - \bar{f}_h'(\bar{z}_h^s) \right)$$

$$= 2 \sum_{k \in [K]} \left( \bar{r}_h^k + \widehat{f}_{h+1}'(\bar{s}_{h+1}^k) - \mathcal{T}_h \widehat{f}_{h+1}'(\bar{z}_h^k) \right) \left( \widetilde{f}_h'(\bar{z}_h^k) - \bar{f}_h'(\bar{z}_h^k) \right)$$

$$+ 2 \sum_{k \in [K]} \left( \mathcal{T}_h \widehat{f}_{h+1}'(\bar{z}_h^k) - \bar{f}_h'(\bar{z}_h^k) \right) \left( \widetilde{f}_h'(\bar{z}_h^k) - \bar{f}_h'(\bar{z}_h^k) \right)$$

$$\leq 2 \sum_{k \in [K]} \left( \bar{r}_h^k + \widehat{f}_{h+1}'(\bar{s}_{h+1}^k) - \mathcal{T}_h \widehat{f}_{h+1}'(\bar{z}_h^k) \right) \left( \widetilde{f}_h'(\bar{z}_h^k) - \bar{f}_h'(\bar{z}_h^k) \right) + 4KL\epsilon$$

$$\leq (24L^2 + 5)i^2(\delta) + 4KL\epsilon + \frac{\sum_{k \in [K]} \left( \widetilde{f}_h'(\bar{z}_h^k) - \mathcal{T}_h \widehat{f}_{h+1}'(\bar{z}_h^k) \right)^2}{2}$$

$$\leq (24L^2 + 5)i^2(\delta) + 8KL\epsilon + \frac{\sum_{k \in [K]} \left( \bar{f}_h'(\bar{z}_h^k) - \widetilde{f}_h'(\bar{z}_h^k) \right)^2}{2}$$

$$\leq \frac{(\beta_{1,h}')^2}{2} + \frac{\sum_{k \in [K]} \left( \bar{f}_h'(\bar{z}_h^k) - \widetilde{f}_h'(\bar{z}_h^k) \right)^2}{2}.$$

568   Here the second inequality holds because of the Bellman completeness assumption (Assumption 3.1).
569   The third inequality arises from (C.5). The last inequality holds due to the choice of

$$\beta_{1,h}' = \sqrt{2(24L^2 + 5)i^2(\delta) + 16KL\epsilon} = \widetilde{O}\left( \sqrt{\log \mathcal{N} \mathcal{N}_b} H \right).$$

570   But conditioned on the event $\mathcal{E}_{1,h}'$, we have

$$\sum_{k \in [K]} \left( \bar{r}_h^k + \widehat{f}_{h+1}'(\bar{s}_{h+1}^k) - \bar{f}_h'(\bar{z}_h^k) \right) \left( \widetilde{f}_h'(\bar{z}_h^k) - \bar{f}_h'(\bar{z}_h^k) \right)$$

$$\geq \sum_{k \in [K]} \left( \bar{f}_h'(\bar{z}_h^k) - \widetilde{f}_h'(\bar{z}_h^k) \right)^2$$

$$> \frac{(\beta_{1,h}')^2}{2} + \frac{\sum_{k \in [K]} \left( \bar{f}_h'(\bar{z}_h^k) - \widetilde{f}_h'(\bar{z}_h^k) \right)^2}{2}.$$

571   Thus, we have $\mathbb{P}[\mathcal{E}_{1,h}'] \leq \delta/(4H^2)$.                                      $\square$

## C.3   Proof of Lemma B.4

573   To prove this lemma, we need a lemma similar to Lemma C.1

574   **Lemma C.2.** On dataset $\mathcal{D}' = \{\bar{s}_h^k, \bar{a}_h^k, \bar{r}_h^k\}_{k,h=1}^{K,H}$, we define the filtration

$$\bar{\mathcal{H}}_h^k = \sigma(\bar{s}_1^1, \bar{a}_1^1, \bar{r}_1^1, \bar{s}_2^1, \dots, \bar{r}_H^1, \bar{s}_{H+1}^1; \bar{s}_1^2, \bar{a}_1^2, \bar{r}_1^2, \bar{s}_2^2, \dots, \bar{r}_H^2, \bar{s}_{H+1}^2; \dots; \bar{s}_1^k, \bar{a}_1^k, \bar{r}_1^k, \bar{s}_2^k, \dots, \bar{r}_h^k, \bar{s}_{h+1}^k).$$

575   For any fixed function $f, f' : \mathcal{S} \to [0, L]$, we make the following definitions:

$$\bar{\eta}_h^k[f'] := \left( \bar{r}_h^k + f'(\bar{s}_{h+1}^k) \right)^2 - \left[ \mathbb{P}_h (\bar{r}_h + f')^2 \right] (\bar{s}_h^k, \bar{a}_h^k)$$

$$\bar{D}_h^k[f, f'] := 2\bar{\eta}_h^k[f'] \left( f(\bar{z}_h^k) - \mathcal{T}_{2,h} f'(\bar{z}_h^k) \right).$$

576   Then with probability at least $1 - \delta/(4H^2 \mathcal{N}^2 \mathcal{N}_b^2)$, the following inequality holds,

$$\sum_{k \in [K]} \bar{D}_h^k[f, f'] \leq (24L + 5)i'^2(\delta) + \frac{\sum_{k \in [K]} \left( f(\bar{z}_h^k) - \mathcal{T}_{2,h} f'(\bar{z}_h^k) \right)^2}{2},$$

577   where $i'(\delta) = \sqrt{4 \log \frac{\mathcal{N} \mathcal{N}_b H (2 \log(4LK) + 2)(\log(4L) + 2)}{\delta}}$.

578 *Proof.* We use Lemma D.1, with the following conditions:

$\bar{D}_h^k[f, f']$ is adapted to the filtration $\bar{\mathcal{H}}_h^k$ and $\mathbb{E}\left[\bar{D}_h^k[f, f'] \mid \bar{\mathcal{H}}_h^{k-1}\right] = 0.$

$$\left|\bar{D}_h^k[f, f']\right| \leq 2|\bar{\eta}_h^k| \max_z |f(z) - \mathcal{T}_{2,h} f'(z)| \leq 4L^2 = M.$$

$$\sum_{k \in [K]} \mathbb{E}\left[\left(\bar{D}_h^k[f, f']\right)^2 \Big| \bar{z}_h^k\right] = \sum_{k \in [K]} \mathbb{E}\left[4(\bar{\eta}_h^k[f'])^2 | \bar{z}_h^k\right] \left(f(\bar{z}_h^k) - \mathcal{T}_{2,h} f'(\bar{z}_h^k)\right)^2 \leq (4L^2 K)^2 = V^2.$$

579 On the other hand,

$$\sum_{k \in [K]} \mathbb{E}\left[\left(\bar{D}_h^k[f, f']\right)^2 \Big| \bar{z}_h^k\right] = \sum_{k \in [K]} \mathbb{E}\left[4\left(\bar{\eta}_h^k[f']\right)^2 \Big| \bar{z}_h^k\right] \left(f(\bar{z}_h^k) - \mathcal{T}_h f'(\bar{z}_h^k)\right)^2$$

$$\leq 8L^4 \sum_{k \in [K]} \left(f(\bar{z}_h^k) - \mathcal{T}_{2,h} f'(\bar{z}_h^k)\right)^2.$$

580 Then using Lemma D.1 with $v = 1$, $m = 1$, we have:

$$\sum_{k \in [K]} 2\bar{\eta}_h^k[f'] \left(f(\bar{z}_h^k) - \mathcal{T}_{2,h} f'(\bar{z}_h^k)\right) \leq i'(\delta) \sqrt{2(2 \cdot 8L^4) \sum_{k \in [K]} \left(f(\bar{z}_h^k) - \mathcal{T}_{2,h} f'(\bar{z}_h^k)\right)^2}$$

$$+ \frac{2}{3} i'^2(\delta) + \frac{4}{3} i'^2(\delta) \cdot 4L^2$$

$$\leq (20L^4 + 5) i'^2(\delta) + \frac{\sum_{k \in [K]} \left(f(\bar{z}_h^k) - \mathcal{T}_h f'(\bar{z}_h^k)\right)^2}{2}$$

581 □

582 *Proof of Lemma B.4.* Let $(\beta'_{2,h})^2 = 2(20L^4 + 5) i'^2(\delta) + 16KL\epsilon$. We define the event $\mathcal{E}'_{2,h} :=$

583 $\left\{\sum_{k \in [K]} \left(\bar{g}'_h(\bar{z}_h^k) - \tilde{g}'_h(\bar{z}_h^k)\right)^2 > (\beta'_{2,h})^2\right\}$. The following inequality will be useful in our proof.

$$\sum_{k \in [K]} \left(\bar{g}'_h(\bar{z}_h^k) - \tilde{g}'_h(\bar{z}_h^k)\right)^2 = \sum_{k \in [K]} \left[\left((\bar{r}_h^k + \widehat{f}'_{h+1}(\bar{s}_{h+1}^k))^2 - \bar{g}'_h(\bar{z}_h^k)\right) + \left(\tilde{g}'_h(\bar{z}_h^k) - (\bar{r}_h^k + \widehat{f}'_{h+1}(\bar{s}_{h+1}^k))^2\right)\right]^2$$

$$= \sum_{k \in [K]} \left((\bar{r}_h^k + \widehat{f}'_{h+1}(\bar{s}_{h+1}^k))^2 - \bar{g}'_h(\bar{z}_h^k)\right)^2 + \sum_{k \in [K]} \left(\tilde{g}'_h(\bar{z}_h^k) - (\bar{r}_h^k + \widehat{f}'_{h+1}(\bar{s}_{h+1}^k))^2\right)^2$$

$$+ 2 \sum_{k \in [K]} \left((\bar{r}_h^k + \widehat{f}'_{h+1}(\bar{s}_{h+1}^k))^2 - \bar{g}'_h(\bar{z}_h^k)\right) \left(\tilde{g}'_h(\bar{z}_h^k) - (\bar{r}_h^k + \widehat{f}'_{h+1}(\bar{s}_{h+1}^k))^2\right)$$

$$\leq 2 \sum_{k \in [K]} \left((\bar{r}_h^k + \widehat{f}'_{h+1}(\bar{s}_{h+1}^k))^2 - \bar{g}'_h(\bar{z}_h^k)\right)^2$$

$$+ 2 \sum_{k \in [K]} \left((\bar{r}_h^k + \widehat{f}'_{h+1}(\bar{s}_{h+1}^k))^2 - \bar{g}'_h(\bar{z}_h^k)\right) \left(\tilde{g}'_h(\bar{z}_h^k) - (\bar{r}_h^k + \widehat{f}'_{h+1}(\bar{s}_{h+1}^k))^2\right)$$

$$\leq 2 \sum_{k \in [K]} \left((\bar{r}_h^k + \widehat{f}'_{h+1}(\bar{s}_{h+1}^k))^2 - \bar{g}'_h(\bar{z}_h^k)\right) \left(\tilde{g}'_h(\bar{z}_h^k) - \bar{g}'_h(\bar{z}_h^k)\right). \quad \text{(C.6)}$$

584 Here we use our choice of $\tilde{g}'_h$, i.e. $\tilde{g}'_h = \arg\min_{g_h \in \mathcal{F}_h} \sum_{k \in [K]} \left(g_h(\bar{s}_h^k, \bar{a}_h^k) - (\bar{r}_h^k + \widehat{f}'_{h+1}(\bar{s}_{h+1}^k))^2\right)^2$.

585 Next, we will use Lemma C.2. For any fixed $h$, let $f = \tilde{g}'_h \in \mathcal{F}_h$, $f' = \widehat{f}'_{h+1} = \{\tilde{f} - \epsilon\}_{[0, H-h+1]}$,

586 where $\tilde{f} = \widehat{f}'_h - b'_h \in \mathcal{F}_h - \mathcal{W}$. Following the construction in Lemma C.1, we define

$$\bar{\eta}_h^k[f'] := \left(\bar{r}_h^k + f'(\bar{s}_{h+1}^k)\right)^2 - \left[\mathbb{P}_h(\bar{r}_h + f')^2\right](\bar{s}_h^k, \bar{a}_h^k)$$

$$\text{and } \bar{D}_h^k[f, f'] := 2\bar{\eta}_h^k[f'] \left(f(\bar{z}_h^k) - \mathcal{T}_{2,h} f'(\bar{z}_h^k)\right).$$

587 Due to the result of Lemma C.2, taking a union bound, we have with probability at least $1 - \delta/(4H^2)$,

588 the following inequality holds,

$$\sum_{k \in [K]} \bar{D}_h^k[f, f'] \leq (20L^4 + 5) i^2(\delta) + \frac{\sum_{k \in [K]} \left(f(\bar{z}_h^k) - \mathcal{T}_{2,h} f'(\bar{z}_h^k)\right)^2}{2}. \quad \text{(C.7)}$$

589 Therefore, with probability at least $1 - \delta/(4H^2)$, we have

$$2 \sum_{k \in [K]} \left( (\bar{r}_h^k + \widehat{f}_{h+1}'(\bar{s}_{h+1}^k))^2 - \bar{g}_h'(\bar{z}_h^k) \right) \left( \widetilde{g}_h'(\bar{z}_h^k) - \bar{g}_h'(\bar{z}_h^k) \right)$$

$$= 2 \sum_{k \in [K]} \left( (\bar{r}_h^k + \widehat{f}_{h+1}'(\bar{s}_{h+1}^k))^2 - \mathcal{T}_{2,h}\widehat{f}_{h+1}'(\bar{z}_h^k) \right) \left( \widetilde{g}_h'(\bar{z}_h^k) - \bar{g}_h'(\bar{z}_h^k) \right)$$

$$+ 2 \sum_{k \in [K]} \left( \mathcal{T}_{2,h}\widehat{f}_{h+1}'(\bar{z}_h^k) - \bar{g}_h'(\bar{z}_h^k) \right) \left( \widetilde{g}_h'(\bar{z}_h^k) - \bar{g}_h'(\bar{z}_h^k) \right)$$

$$\leq 2 \sum_{k \in [K]} \left( (\bar{r}_h^k + \widehat{f}_{h+1}'(\bar{s}_{h+1}^k))^2 - \mathcal{T}_{2,h}\widehat{f}_{h+1}'(\bar{z}_h^k) \right) \left( \widetilde{g}_h'(\bar{z}_h^k) - \bar{g}_h'(\bar{z}_h^k) \right) + 4KL\epsilon$$

$$\leq (20L^4 + 5)i'^2(\delta) + 4KL\epsilon + \frac{\sum_{k \in [K]} \left( \bar{g}_h'(\bar{z}_h^k) - \mathcal{T}_{2,h}\widehat{f}_{h+1}'(\bar{z}_h^k) \right)^2}{2}$$

$$\leq (20L^4 + 5)i'^2(\delta) + 8KL\epsilon + \frac{\sum_{k \in [K]} \left( \bar{g}_h'(\bar{z}_h^k) - \widetilde{g}_h'(\bar{z}_h^k) \right)^2}{2}$$

$$\leq \frac{(\beta_{2,h}')^2}{2} + \frac{\sum_{k \in [K]} \left( \bar{g}_h'(\bar{z}_h^k) - \widetilde{g}_h'(\bar{z}_h^k) \right)^2}{2}.$$

590 Here the second inequality holds because of the Bellman completeness assumption (Assumption 3.1).
591 The third inequality arises from (C.7). The last inequality holds due to the choice of

$$\beta_{2,h}' = \sqrt{2(20L^4 + 5)i'^2(\delta) + 16KL\epsilon} = \widetilde{O}(\sqrt{\log \mathcal{N}\mathcal{N}_b}H^2).$$

592 But conditioned on the event $\mathcal{E}_{2,h}'$, we have

$$\sum_{k \in [K]} \left( (\bar{r}_h^k + \widehat{f}_{h+1}'(\bar{s}_{h+1}^k))^2 - \bar{g}_h'(\bar{z}_h^k) \right) \left( \widetilde{g}_h'(\bar{z}_h^k) - \bar{g}_h'(\bar{z}_h^k) \right)$$

$$\geq \sum_{k \in [K]} \left( \bar{g}_h'(\bar{z}_h^k) - \widetilde{g}_h'(\bar{z}_h^k) \right)^2$$

$$> \frac{(\beta_{2,h}')^2}{2} + \frac{\sum_{k \in [K]} \left( \bar{g}_h'(\bar{z}_h^k) - \widetilde{g}_h'(\bar{z}_h^k) \right)^2}{2}.$$

593 Here we use (C.6). Thus, we have $\mathbb{P}[\mathcal{E}_{2,h}'] \leq \delta/(4H^2)$. $\qquad\square$

594 ## C.4  Proof of Lemma B.5

595 *Proof of Lemma B.5.* We write $\mathbb{B}_h(s, a) = \widetilde{g}_h'(s, a) - \left( \widetilde{f}_h'(s, a) \right)^2$. We first bound the difference
596 between $\mathbb{B}_h(s, a)$ and $[\text{Var}_h \widehat{f}_{h+1}'](s, a)$. By the definition of conditional variance, we have

$$\left| \mathbb{B}_h(s, a) - [\text{Var}_h \widehat{f}_{h+1}'](s, a) \right| \leq \left| \widetilde{g}_h'(s, a) - \mathcal{T}_{2,h}\widehat{f}_{h+1}'(s, a) \right| + \left| \left( \widetilde{f}_h'(s, a) \right)^2 - \left( \mathcal{T}_h \widehat{f}_{h+1}'(s, a) \right)^2 \right|,$$

597 where we use our definition of Bellman operators. By the Bellman completeness assumption, there
598 exists $\bar{f}_h' \in \mathcal{F}_h$, $\bar{g}_h' \in \mathcal{F}_h$, such that $\left| \bar{f}_h'(s, a) - \mathcal{T}_h \widehat{f}_{h+1}'(s, a) \right| \leq \epsilon$, $\left| \bar{g}_h'(s, a) - \mathcal{T}_{2,h}\widehat{f}_{h+1}'(s, a) \right| \leq \epsilon$
599 for all $(s, a)$. Then by Lemma B.3 we can see that with probability at least $1 - \delta/(4H^2)$, the following
600 inequality holds

$$\sum_{k \in [K]} \left( \bar{f}_h'(\bar{z}_h^k) - \widetilde{f}_h'(\bar{z}_h^k) \right)^2 \leq (\beta_{1,h}')^2. \tag{C.8}$$

601 Similarly, for the second order term, using Lemma B.4, we can see that with probability at least
602 $1 - \delta/(4H^2)$, the following inequality holds

$$\sum_{k \in [K]} \left( \bar{g}_h'(\bar{z}_h^k) - \widetilde{g}_h'(\bar{z}_h^k) \right)^2 \leq (\beta_{2,h}')^2. \tag{C.9}$$

After taking a union bound, we have that with probability at least $1 - \delta/(2H)$, (C.8) and (C.9) hold for all $h \in [H]$ simultaneously. Under this high-probability event, we have

$$
\left| \widetilde{g}'_h(s,a) - \mathcal{T}_{2,h}\widehat{f}'_{h+1}(s,a) \right| + \left| \left(\widetilde{f}_h(s,a)\right)^2 - \left(\mathcal{T}_h\widehat{f}'_{h+1}(s,a)\right)^2 \right|
$$

$$
\leq \epsilon + |\widetilde{g}'_h(s,a) - \bar{g}'_h(s,a)| + O(H) \cdot \left| \widetilde{f}'_h(s,a) - \bar{f}'_h(s,a) + \epsilon \right|
$$

$$
\leq O(H) \cdot \epsilon + \frac{|\widetilde{g}'_h(s,a) - \bar{g}'_h(s,a)|}{\sqrt{\sum_{k\in[K]} \left(\bar{g}'_h(\bar{z}^k_h) - \widetilde{g}'_h(\bar{z}^k_h)\right)^2 + \lambda}} \cdot \sqrt{\sum_{k\in[K]} \left(\bar{g}'_h(\bar{z}^k_h) - \widetilde{g}'_h(\bar{z}^k_h)\right)^2 + \lambda}
$$

$$
+ O(H) \cdot \frac{|\widetilde{f}'_h(s,a) - \bar{f}'_h(s,a)|}{\sqrt{\sum_{k\in[K]} \left(\bar{f}'_h(\bar{z}^k_h) - \widetilde{f}'_h(\bar{z}^k_h)\right)^2 + \lambda}} \cdot \sqrt{\sum_{k\in[K]} \left(\bar{f}'_h(\bar{z}^k_h) - \widetilde{f}'_h(\bar{z}^k_h)\right)^2 + \lambda}
$$

$$
\leq O(H) \cdot \epsilon + \frac{|\widetilde{g}'_h(s,a) - \bar{g}'_h(s,a)|}{\sqrt{\sum_{k\in[K]} \left(\bar{g}'_h(\bar{z}^k_h) - \widetilde{g}'_h(\bar{z}^k_h)\right)^2 + \lambda}} \cdot \sqrt{(\beta'_{2,h})^2 + \lambda}
$$

$$
+ O(H) \cdot \frac{|\widetilde{f}'_h(s,a) - \bar{f}'_h(s,a)|}{\sqrt{\sum_{k\in[K]} \left(\bar{f}'_h(\bar{z}^k_h) - \widetilde{f}'_h(\bar{z}^k_h)\right)^2 + \lambda}} \cdot \sqrt{(\beta'_{1,h})^2 + \lambda}
$$

$$
\leq \widetilde{O}(\sqrt{\log \mathcal{N}\mathcal{N}_b} H^2) \cdot D_{\mathcal{F}_h}(z, \mathcal{D}'_h, 1)
$$

$$
\leq \widetilde{O}\left(\frac{\sqrt{\log \mathcal{N}\mathcal{N}_b} H^2}{\sqrt{K\kappa}}\right),
$$

where the first inequality holds due to the completeness assumption. The third inequality holds because of (C.8) and (C.9). The fourth inequality holds due to the definition of $D^2$-divergence. Finally we use Lemma B.1.

To further bound the difference between $\left[\mathrm{Var}_h \widehat{f}'_{h+1}\right](s,a)$ and $[\mathrm{Var}_h V^*_{h+1}](s,a)$ under the event when (C.8) and (C.9) hold for all $h \in [H]$ simultaneously, we first prove $\|\widehat{f}'_{h+1} - V^*_{h+1}\|_\infty \leq \widetilde{O}\left(\frac{\sqrt{\log \mathcal{N}\mathcal{N}_b} H^3}{\sqrt{K\kappa}}\right)$ by induction.

At stage $H + 1$, $\widehat{f}'_{H+1} = V^*_{H+1} = 0$, the inequality holds naturally. At stage $H$, we have

$$
\begin{aligned}
Q^*_H(s,a) &= \mathcal{T}_H V^*_{H+1}(s,a) \\
&= \mathcal{T}_H \widehat{f}'_{H+1}(s,a) \\
&\geq \widetilde{f}'_H(s,a) - |\mathcal{T}_H \widehat{f}'_{H+1}(s,a) - \widetilde{f}'_H(s,a)| \\
&\geq \widetilde{f}'_H(s,a) - (\epsilon + |\bar{f}'_H(s,a) - \widetilde{f}'_H(s,a)|) \\
&\geq \widetilde{f}'_H(s,a) - b'_H(s,a) - \epsilon \\
&= \widehat{f}'_H(s,a).
\end{aligned}
$$

Here we use the definition of $\widehat{f}'_H$ in Algorithm 1 Line 6. Lemma B.3 shows

$$
\sum_{k\in[K]} \left(\bar{f}'_h(\bar{z}^k_h) - \widetilde{f}'_h(\bar{z}^k_h)\right)^2 \leq (\beta'_{1,h})^2.
$$

Then the fifth inequality follows the bonus oracle assumption (Definition 4.1). Therefore, $V^*_H(s) \geq \widehat{f}'_H(s)$ for all $s \in \mathcal{S}$.

We also have

$$
\begin{aligned}
V_H^*(s) - \widehat{f}_H'(s) &= \langle Q_H^*(s, \cdot) - \widehat{f}_H'(s, \cdot), \pi^*(\cdot|s)\rangle_{\mathcal{A}} + \langle \widehat{f}_H'(s, \cdot), \pi_H^*(\cdot|s) - \widehat{\pi}_H(\cdot|s)\rangle_{\mathcal{A}} \\
&\leq \langle Q_H^*(s, \cdot) - \widehat{f}_H'(s, \cdot), \pi^*(\cdot|s)\rangle_{\mathcal{A}} \\
&= \langle \mathcal{T}_H V_H^*(s, \cdot) - \widetilde{f}_H'(s, \cdot) + b_H'(s, \cdot), \pi^*(\cdot|s)\rangle_{\mathcal{A}} \\
&= \langle \mathcal{T}_H \widehat{f}_{H+1}'(s, \cdot) - \widetilde{f}_H'(s, \cdot) + b_H'(s, \cdot), \pi^*(\cdot|s)\rangle_{\mathcal{A}} \\
&\quad + \langle \mathcal{T}_H V_{H+1}^*(s, \cdot) - \mathcal{T}_H \widehat{f}_{H+1}'(s, \cdot), \pi_H^*(\cdot|s)\rangle_{\mathcal{A}} \\
&\leq 2\langle b_H'(s, \cdot), \pi_H^*(\cdot, s)\rangle_{\mathcal{A}} + \epsilon \\
&\leq \widetilde{O}\left(\frac{\sqrt{\log \mathcal{N}\mathcal{N}_b} H}{\sqrt{K\kappa}}\right),
\end{aligned}
$$

where the second inequality holds due to the selection of policy $\widehat{\pi}_H$. In the fifth inequality, we use the Bellman completeness assumption:

$$
\left|\bar{f}_H'(z) - \mathcal{T}_H \widehat{f}_{H+1}'(z)\right| \leq \epsilon, \forall z \in \mathcal{S} \times \mathcal{A}.
$$

and Lemma B.3. The last inequality holds because of Definition 4.1 and Lemma B.1.

We define $R_h = \widetilde{O}\left(\frac{\sqrt{\log \mathcal{N}\mathcal{N}_b} H}{\sqrt{K\kappa}}\right) \cdot (H - h + 1)$. To use the method of induction, we define the induction assumption as follows: Suppose with the probability of $1 - \delta_{h+1}$, the event $\mathcal{E}_{h+1} = \{0 \leq V_{h+1}^*(s) - \widehat{V}_{h+1}'(s) \leq R_{h+1}\}$ holds. Then we want to prove that with the probability of $1 - \delta_h$, the event $\mathcal{E}_h = \{0 \leq V_h^*(s) - \widehat{V}_h'(s) \leq R_h\}$ holds.

Conditioned on the event $\mathcal{E}_{h+1}$, using similar argument to stage $H$, we have

$$
\begin{aligned}
Q_h^*(s, a) &= \mathcal{T}_h V_{h+1}^*(s, a) \\
&\geq \mathcal{T}_h \widehat{f}_{h+1}'(s, a) \\
&\geq \widetilde{f}_h'(s, a) - |\mathcal{T}_h \widehat{f}_{h+1}'(s, a) - \widetilde{f}_h'(s, a)| \\
&\geq \widetilde{f}_h'(s, a) - \left(\epsilon + |\bar{f}_h'(s, a) - \widetilde{f}_h'(s, a)|\right) \\
&\geq \widetilde{f}_h'(s, a) - b_h'(s, a) \\
&= \widehat{f}_h'(s, a).
\end{aligned}
$$

Therefore, $V_h^*(\cdot) \geq \widehat{f}_h'(\cdot)$.

On the other hand, similar to the case at stage $H$, we have with probability at least $1 - \delta_h - \delta/(2H^2)$,

$$
\begin{aligned}
V_h^*(s) - \widehat{f}_h'(s) &= \langle Q_h^*(s, \cdot) - \widehat{f}_h'(s, \cdot), \pi_h^*(\cdot|s)\rangle_{\mathcal{A}} + \langle \widehat{f}_h'(s, \cdot), \pi_h^*(\cdot|s) - \widehat{\pi}_h(\cdot|s)\rangle_{\mathcal{A}} \\
&\leq \langle Q_h^*(s, \cdot) - \widehat{f}_h'(s, \cdot), \pi_h^*(\cdot|s)\rangle_{\mathcal{A}} \\
&= \langle \mathcal{T}_h V_{h+1}^*(s, \cdot) - \widetilde{f}_h'(s, \cdot) + b_h'(s, a), \pi_h^*(\cdot|s)\rangle_{\mathcal{A}} \\
&= \langle \mathcal{T}_h \widehat{f}_{h+1}'(s, \cdot) - \widetilde{f}_h'(s, \cdot) + b_h'(s, a), \pi_h^*(\cdot|s)\rangle_{\mathcal{A}} \\
&\quad + \langle \mathcal{T}_h V_{h+1}^*(s, \cdot) - \mathcal{T}_h \widehat{f}_{h+1}'(s, \cdot), \pi_h^*(\cdot|s)\rangle_{\mathcal{A}} \\
&\leq 2\langle b_h'(s, \cdot), \pi_h^*(\cdot, s)\rangle_{\mathcal{A}} + \epsilon + R_{h+1} \\
&\leq R_{h+1} + \widetilde{O}\left(\frac{\sqrt{\log \mathcal{N}\mathcal{N}_b} H}{\sqrt{K\kappa}}\right) \\
&\leq \widetilde{O}\left(\frac{\sqrt{\log \mathcal{N}\mathcal{N}_b} H}{\sqrt{K\kappa}}\right) \cdot (H - h + 1) = R_h.
\end{aligned}
$$

The induction shows we can choose $\delta_h = h\delta/(2H^2)$. Thus, taking a union bound over all $h \in [H]$, we prove that with probability at least $1 - \delta/2$, the following inequality

$$
0 \leq V_{h+1}^*(\cdot) - \widehat{f}_{h+1}'(\cdot) \leq \widetilde{O}\left(\frac{\sqrt{\log \mathcal{N}\mathcal{N}_b} H^2}{\sqrt{K\kappa}}\right) \tag{C.10}
$$

628    holds for all $h \in [H]$ simultaneously.

629    Conditioned on this event, we can further bound the difference between $[\text{Var}_h \widehat{f}'_{h+1}](s,a)$ and
630    $[\text{Var}_h V^*_{h+1}](s,a)$.

$$
\begin{aligned}
\left| [\text{Var}_h \widehat{f}'_{h+1}](s,a) - [\text{Var}_h V^*_{h+1}](s,a) \right| &\leq \left| [\mathbb{P}_h \widehat{f}'^2_{h+1}](s,a) - [\mathbb{P}_h V^{*2}_{h+1}](s,a) \right| \\
&\quad + \left| \left( [\mathbb{P}_h \widehat{f}'_{h+1}](s,a) \right)^2 - \left( [\mathbb{P}_h V^*_{h+1}](s,a) \right)^2 \right| \\
&\leq O(H) \cdot \left\| V^*_{h+1} - \widehat{f}'_{h+1} \right\|_\infty \\
&\leq \widetilde{O} \left( \frac{\sqrt{\log \mathcal{N} \mathcal{N}_b} H^3}{\sqrt{K \kappa}} \right).
\end{aligned}
$$

631    The last inequality arises from (C.10). Therefore, for any $(s,a) \in \mathcal{S} \times \mathcal{A}$, we have

$$
\begin{aligned}
\left| \mathbb{B}_h(s,a) - [\text{Var}_h V^*_{h+1}](s,a) \right| &\leq \left| \mathbb{B}_h(s,a) - [\text{Var}_h \widehat{f}'_{h+1}](s,a) \right| \\
&\quad + \left| [\text{Var}_h \widehat{f}'_{h+1}](s,a) - [\text{Var}_h V^*_{h+1}](s,a) \right| \\
&\leq \widetilde{O} \left( \frac{\sqrt{\log \mathcal{N} \mathcal{N}_b} H^3}{\sqrt{K \kappa}} \right).
\end{aligned}
$$

632    Thus, for any $(s,a) \in \mathcal{S} \times \mathcal{A}$, we have

$$
\mathbb{B}_h(s,a) - \widetilde{O} \left( \frac{\sqrt{\log \mathcal{N} \mathcal{N}_b} H^3}{\sqrt{K \kappa}} \right) \leq [\text{Var}_h V^*_{h+1}](s,a).
$$

633    Finally, using the fact that the function $\max\{1, \cdot\}$ is increasing and nonexpansive, we finish the proof
634    of Lemma B.5, which is

$$
[\mathbb{V}_h V^*_{h+1}](s,a) - \widetilde{O} \left( \frac{\sqrt{\log \mathcal{N} \mathcal{N}_b} H^3}{\sqrt{K \kappa}} \right) \leq \widehat{\sigma}^2_h(s,a) \leq [\mathbb{V}_h V^*_{h+1}](s,a).
$$

635    $\hfill\square$

## C.5    Proof of Lemma B.6

637    To prove this result, we need the following lemmas.

638    **Lemma C.3.** Based on the dataset $\mathcal{D} = \{s^k_h, a^k_h, r^k_h\}^{K,H}_{k,h=1}$, we define the filtration $\mathcal{H}^k_h =$
639    $\sigma(s^1_1, a^1_1, r^1_1, s^1_2, \ldots, r^1_H, s^1_{H+1}; x^2_1, a^2_1, r^2_1, s^2_2, \ldots, r^2_H, s^2_{H+1}; \cdots s^k_1, a^k_1, r^k_1, s^k_2, \ldots, r^k_h, s^k_{h+1})$. For
640    any fixed function $f, f' : \mathcal{S} \to \in [0, L]$, we define the following random variables:

$$
\eta^k_h := V^*_{h+1}(s^k_{h+1}) - [\mathbb{P}_h V^*_{h+1}](s^k_h, a^k_h)
$$

$$
D^s_h[f, f'] := 2 \frac{\eta^k_h}{(\widehat{\sigma}_h(z^k_h))^2} \left( f(z^k_h) - f'(z^k_h) \right),
$$

641    Suppose the variance function $\widehat{\sigma}_h$ satisfies the inequality in Lemma B.5, where

$$
[\mathbb{V}_h V^*_{h+1}](s,a) - \widetilde{O} \left( \frac{\sqrt{\log \mathcal{N} \mathcal{N}_b} H^3}{\sqrt{K \kappa}} \right) \leq \widehat{\sigma}^2_h(s,a) \leq [\mathbb{V}_h V^*_{h+1}](s,a).
$$

642    Then, with probability at least $1 - \delta/(4H^2 \mathcal{N}^2)$, the following inequality holds,

$$
\sum_{k \in [K]} D^k_h[f, f'] \leq \frac{4}{3} v(\delta) \sqrt{\lambda} + \sqrt{2} v(\delta) + 30 v^2(\delta) + \frac{1}{v(\delta)} \sqrt{\sum_{k \in [K]} \frac{1}{(\widehat{\sigma}_h(z^k_h))^2} \left( f(z^k_h) - f'(z^k_h) \right)^2 + \lambda},
$$

643    where $v(\delta) = \sqrt{2 \log \frac{H \mathcal{N} (2 \log(18LT) + 2)(\log(18L) + 2)}{\delta_h}}$.

644   *Proof.* We use Lemma D.1, with the following conditions:

$$D_h^k[f, f'] \text{ is adapted to the filtration } \mathcal{H}_h^k \text{ and } \mathbb{E}\left[D_h^k[f, f'] \mid \mathcal{H}_h^{k-1}\right] = 0.$$

$$\left|D_h^k[f, f']\right| \leq 2\left|\eta_h^k\right| \max_z |f(z) - f'(z)| \leq 8LH = M.$$

$$\sum_{k \in [K]} \mathbb{E}\left[\left(D_h^k[f, f']\right)^2 \Big| z_h^k\right] = 4 \sum_{k \in [K]} \frac{\mathbb{E}\left[(\eta_h^k)^2 | z_h^k\right]}{(\widehat{\sigma}_h(z_h^k))^4} \left(f(z_h^k) - f'(z_h^k)\right).$$

645   On the other hand,

$$\sum_{k \in [K]} \mathbb{E}\left[\left(D_h^k[f, f']\right)^2 \Big| z_h^k\right] = 4 \sum_{k \in [K]} \frac{\mathbb{E}\left[(\eta_h^k)^2 | z_h^k\right]}{(\widehat{\sigma}_h(z_h^k))^4} \left(f(z_h^k) - f'(z_h^k)\right)^2$$

$$\leq 8 \sum_{k \in [K]} \frac{1}{(\widehat{\sigma}_h(z_h^k))^2} \left(f(z_h^k) - f'(z_h^k)\right)^2,$$

646   where the last inequality holds because of the inequality in Lemma B.5:

$$\mathbb{E}\left[(\eta_h^k)^2 | z_h^k\right] = [\mathrm{Var}_h V_{h+1}^*](s_h^k, a_h^k)$$

$$\leq [\mathbb{V}_h V_{h+1}^*](s_h^k, a_h^k)$$

$$\leq \left(\widehat{\sigma}_h(z_h^k)\right)^2 + \widetilde{O}\left(\frac{\sqrt{\log \mathcal{N}\mathcal{N}_b} H^3}{\sqrt{K\kappa}}\right)$$

$$\leq 2\left(\widehat{\sigma}_h(z_h^k)\right)^2,$$

647   where we use the requirement that $K \geq \widetilde{\Omega}\left(\frac{\log \mathcal{N}\mathcal{N}_b H^6}{\kappa}\right)$.

648   Moreover, for any $k \in [K]$,

$$\left|D_h^k[f, f']\right| \leq 2\left|\frac{\eta_h^k}{(\widehat{\sigma}_h(z_h^k))^2}\right| \left|f(z_h^k) - f'(z_h^k)\right|$$

$$\leq 4H \sqrt{D_{\mathcal{F}_h}^2(z_h^k, \mathcal{D}_h, \widehat{\sigma}_h)\left(\sum_{k \in [K]} \frac{1}{(\widehat{\sigma}_h(z_h^k))^2}\left(f(z_h^k) - f'(z_h^k)\right)^2 + \lambda\right)}$$

$$\leq \widetilde{O}\left(\frac{4H^2}{\sqrt{K\kappa}}\right) \sqrt{\sum_{k \in [K]} \frac{1}{(\widehat{\sigma}_h(z_h^k))^2}\left(f(z_h^k) - f'(z_h^k)\right)^2 + \lambda}$$

$$\leq \frac{1}{\upsilon(\delta)} \sqrt{\sum_{k \in [K]} \frac{1}{(\widehat{\sigma}_h(z_h^k))^2}\left(f(z_h^k) - f'(z_h^k)\right)^2 + \lambda}.$$

649   The second inequality holds because of the definition of $D^2$ divergence (Definition 3.2). The
650   third inequality holds due to Lemma B.2. The last inequality holds because of the choice of
651   $K \geq \widetilde{\Omega}\left(\frac{\upsilon^2(\delta) H^4}{\kappa}\right)$.

652   Then using Lemma D.1 with $\upsilon = 1$, $m = 1$, we have

$$\sum_{k \in [K]} 2\frac{\eta_h^k}{(\widehat{\sigma}_h(z_h^k))^2}\left(f(z_h^k) - f'(z_h^k)\right) \leq \upsilon(\delta)\sqrt{16 \sum_{k \in [K]} \frac{1}{(\widehat{\sigma}_h(z_h^k))^2}\left(f(z_h^k) - f'(z_h^k)\right)^2 + 2} + \frac{2}{3}\upsilon^2(\delta)$$

$$+ \frac{4}{3}\upsilon(\delta)\sqrt{\sum_{k \in [K]} \frac{1}{(\widehat{\sigma}_h(z_h^k))^2}\left(f(z_h^k) - f'(z_h^k)\right)^2 + \lambda}$$

$$\leq \frac{4}{3}\upsilon(\delta)\sqrt{\lambda} + \sqrt{2}\upsilon(\delta) + 30\upsilon^2(\delta)$$

$$+ \frac{\sum_{k \in [K]} \frac{1}{(\widehat{\sigma}_h(z_h^k))^2}\left(f(z_h^k) - f'(z_h^k)\right)^2}{4}.$$

653 $\qquad\qquad\qquad\qquad\qquad\qquad\qquad\qquad\qquad\qquad\qquad\qquad\qquad\qquad\qquad\qquad\qquad$ $\square$

**Lemma C.4.** Based on the dataset $\mathcal{D} = \left\{ s_h^k, a_h^k, r_h^k \right\}_{k,h=1}^{K,H}$, we define the following filtration $\mathcal{H}_h^k = \sigma \left( s_1^1, a_1^1, r_1^1, s_2^1, \ldots, r_H^1, s_{H+1}^1; x_1^2, a_1^2, r_1^2, s_2^2, \ldots, r_H^2, s_{H+1}^2; \cdots s_1^k, a_1^k, r_1^k, s_2^k, \ldots, r_h^k, s_{h+1}^k \right)$. For any fixed functions $f, \widetilde{f} : \mathcal{S} \to [0, L]$ and $f' : \mathcal{S} \to [0, H]$, we define the following random variables

$$\xi_h^k[f'] := f'(s_{h+1}^k) - V_{h+1}^*(s_{h+1}^k) - \left[ \mathbb{P}_h(f' - V_{h+1}^*) \right](s_h^k, a_h^k),$$

$$\Delta_h^k \left[ f, \widetilde{f}, f' \right] := 2 \frac{\xi_h^k[f']}{(\widehat{\sigma}_h(z_h^k))^2} \left( f(z_h^k) - \widetilde{f}(z_h^k) \right),$$

Suppose the variance function $\widehat{\sigma}_h$ satisfies the inequality in Lemma B.5, where

$$[\mathbb{V}_h V_{h+1}^*](s, a) - \widetilde{O} \left( \frac{\sqrt{\log \mathcal{N} \mathcal{N}_b} H^3}{\sqrt{K\kappa}} \right) \le \widehat{\sigma}_h^2(s, a) \le [\mathbb{V}_h V_{h+1}^*](s, a).$$

Then, with probability at least $1 - \delta/(4H^2 \mathcal{N}^3 \mathcal{N}_b)$, the following inequality holds,

$$\sum_{k \in [K]} \Delta_h^k[f, \widetilde{f}, f'] \le \left( \frac{4}{3} \iota(\delta) \sqrt{\lambda} + \sqrt{2} \iota(\delta) \right) \| f' - V_{h+1}^* \|_\infty^2 + \frac{2}{3} \iota^2(\delta) / \log \mathcal{N}_b$$

$$+ 30 \iota^2(\delta) \| f' - V_{h+1}^* \|_\infty^2 + \frac{\sum_{k \in [K]} \frac{1}{(\widehat{\sigma}_h(z_h^k))^2} (f(z_h^k) - f'(z_h^k))^2}{4}.$$

where $\iota(\delta) = \sqrt{3 \log \frac{H \mathcal{N} \mathcal{N}_b (2 \log(18LT) + 2)(\log(18L) + 2)}{\delta}}$.

*Proof.* $\Delta_h^k[f, \widetilde{f}, f']$ is adapted to the filtration $\mathcal{H}_h^k$ and $\mathbb{E} \left[ \Delta_h^k[f, \widetilde{f}, f'] \mid \mathcal{H}_h^{k-1} \right] = 0$. We also have

$$\sum_{k \in [K]} \mathbb{E} \left[ (\Delta_h^k[f, \widetilde{f}, f'])^2 \Big| z_h^k \right] = 4 \sum_{k \in [K]} \frac{\mathbb{E} \left[ (\xi_h^k[f'])^2 \Big| z_h^k \right]}{(\widehat{\sigma}_h(z_h^k))^4} \left( f(z_h^k) - f'(z_h^k) \right)^2$$

$$\le 8 \sum_{k \in [K]} \frac{\| f' - V_{h+1}^* \|_\infty^2}{(\widehat{\sigma}_h(z_h^k))^2} \left( f(z_h^k) - f'(z_h^k) \right)^2.$$

Moreover, for any $k \in [K]$,

$$\left| \Delta_h^k[f, \widetilde{f}, f'] \right| \le 2 \left| \frac{\xi_h^k[f']}{(\widehat{\sigma}_h(z_h^k))^2} \right| \left| f(z_h^k) - f'(z_h^k) \right|$$

$$\le 4 \| f' - V_{h+1}^* \|_\infty \sqrt{D_{\mathcal{F}_h}^2(z_h^k, \mathcal{D}_h, \widehat{\sigma}_h) \left( \sum_{k \in [K]} \frac{1}{(\widehat{\sigma}_h(z_h^k))^2} \left( f(z_h^k) - f'(z_h^k) \right)^2 + \lambda \right)}$$

$$\le \widetilde{O} \left( \frac{H}{\sqrt{K\kappa}} \right) \cdot \| f' - V_{h+1}^* \|_\infty \sqrt{\sum_{k \in [K]} \frac{1}{(\widehat{\sigma}_h(z_h^k))^2} \left( f(z_h^k) - f'(z_h^k) \right)^2 + \lambda}$$

$$\le \frac{\| f' - V_{h+1}^* \|_\infty}{\iota(\delta)} \sqrt{\sum_{k \in [K]} \frac{1}{(\widehat{\sigma}_h(z_h^k))^2} \left( f(z_h^k) - f'(z_h^k) \right)^2 + \lambda}$$

The second inequality holds because of the definition of $D^2$ divergence (Definition 3.2). The third inequality holds due to Lemma B.2. The last inequality holds because of the choice of $K \ge \widetilde{\Omega} \left( \frac{\iota^2(\delta) H^4}{\kappa} \right)$.

Then using Lemma D.1 with $v = 1$, $m = 1/\log \mathcal{N}_b$, we have

$$\sum_{k \in [K]} 2 \frac{\xi_h^k[f']}{(\widehat{\sigma}_h(z_h^k))^2} \left( f(z_h^k) - \widetilde{f}(z_h^k) \right) \leq \iota(\delta) \sqrt{8 \sum_{k \in [K]} \frac{\|f' - V_{h+1}^*\|_\infty^2}{(\widehat{\sigma}_h(z_h^k))^2} \left( f(z_h^k) - f'(z_h^k) \right)^2 + 2}$$

$$+ \frac{2}{3} \iota^2(\delta) / \log \mathcal{N}_b + \frac{4}{3} \iota(\delta) \|f' - V_{h+1}^*\|_\infty \sqrt{\sum_{k \in [K]} \frac{1}{(\widehat{\sigma}_h(z_h^k))^2} (f(z_h^k) - f'(z_h^k))^2 + \lambda}$$

$$\leq \left( \frac{4}{3} \iota(\delta) \sqrt{\lambda} + \sqrt{2} \iota(\delta) \right) \|f' - V_{h+1}^*\|_\infty^2 + \frac{2}{3} \iota^2(\delta) / \log \mathcal{N}_b + 30 \iota^2(\delta) \|f' - V_{h+1}^*\|_\infty^2$$

$$+ \frac{\sum_{k \in [K]} \frac{1}{(\widehat{\sigma}_h(z_h^k))^2} \left( f(z_h^k) - f'(z_h^k) \right)^2}{4}.$$

$\square$

*Proof of Lemma B.6.* We define the event $\mathcal{E}_h := \left\{ \sum_{k \in [K]} \frac{1}{(\widehat{\sigma}(z_h^k))^2} \left( \bar{f}_h(z_h^k) - \widetilde{f}_h(z_h^k) \right)^2 > (\beta_h)^2 \right\}$.

The following inequality will be useful in our proof.

$$\sum_{k \in [K]} \frac{1}{(\widehat{\sigma}(z_h^k))^2} \left( \bar{f}_h(z_h^k) - \widetilde{f}_h(z_h^k) \right)^2$$

$$= \sum_{k \in [K]} \frac{1}{(\widehat{\sigma}(z_h^k))^2} \left[ \left( r_h^k + \widehat{f}_{h+1}(s_{h+1}^k) - \bar{f}_h(z_h^k) \right) + \left( \widetilde{f}_h(z_h^k) - r_h^k - \widehat{f}_{h+1}(s_{h+1}^k) \right) \right]^2$$

$$= \sum_{k \in [K]} \frac{1}{(\widehat{\sigma}(z_h^k))^2} \left( r_h^k + \widehat{f}_{h+1}(s_{h+1}^k) - \bar{f}_h(z_h^k) \right)^2 + \sum_{k \in [K]} \frac{1}{(\widehat{\sigma}(z_h^k))^2} \left( \widetilde{f}_h(z_h^k) - r_h^k - \widehat{f}_{h+1}(s_{h+1}^k) \right)^2$$

$$+ 2 \sum_{k \in [K]} \frac{1}{(\widehat{\sigma}(z_h^k))^2} \left( r_h^k + \widehat{f}_{h+1}(s_{h+1}^k) - \bar{f}_h(z_h^k) \right) \left( \widetilde{f}_h(z_h^k) - r_h^k - \widehat{f}_{h+1}(s_{h+1}^k) \right)$$

$$\leq 2 \sum_{k \in [K]} \frac{1}{(\widehat{\sigma}(z_h^k))^2} \left( r_h^k + \widehat{f}_{h+1}(s_{h+1}^k) - \bar{f}_h(z_h^k) \right)^2$$

$$+ 2 \sum_{k \in [K]} \frac{1}{(\widehat{\sigma}(z_h^k))^2} \left( r_h^k + \widehat{f}_{h+1}(s_{h+1}^k) - \bar{f}_h(z_h^k) \right) \left( \widetilde{f}_h(z_h^k) - r_h^k - \widehat{f}_{h+1}(s_{h+1}^k) \right)$$

$$\leq 2 \sum_{k \in [K]} \frac{1}{(\widehat{\sigma}(z_h^k))^2} \left( r_h^k + \widehat{f}_{h+1}(s_{h+1}^k) - \bar{f}_h(z_h^k) \right) \left( \widetilde{f}_h(z_h^k) - \bar{f}_h(z_h^k) \right). \tag{C.11}$$

In the third inequality, we use our choice of $\widetilde{f}_h$ in Algorithm 1 Line 10,

$$\widetilde{f}_h = \underset{f_h \in \mathcal{F}_h}{\arg\min} \sum_{k \in [K]} \frac{1}{(\widehat{\sigma}(z_h^k))^2} \left( f_h(s_h^k, a_h^k) - r_h^k - \widehat{f}_{h+1}(s_{h+1}^k) \right)^2.$$

We first use Lemma C.3 at stage $H$. Let $f = \widetilde{f}_H \in \mathcal{F}_H$, $f' = \bar{f}_H \in \mathcal{F}_H$. We define

$$\eta_H^k := V_{H+1}^*(s_{H+1}^k) - [\mathbb{P}_H V_{H+1}^*](z_H^k)$$

$$D_H^k[f, f'] := 2 \frac{\eta_H^k}{(\widehat{\sigma}_H(z_H^k))^2} \left( f(z_H^k) - f'(z_H^k) \right).$$

Taking a union bound, we have with probability at least $1 - \delta/(4H^2)$, the following inequality holds,

$$\sum_{k \in [K]} 2 \frac{\eta_H^k}{(\widehat{\sigma}_H(z_H^k))^2} \left( \widetilde{f}(z_H^k) - \bar{f}(z_H^k) \right) \leq \frac{4}{3} v(\delta) \sqrt{\lambda} + \sqrt{2} v(\delta) + 30 v^2(\delta)$$

$$+ \frac{\sum_{k \in [K]} \frac{1}{(\widehat{\sigma}_H(z_H^k))^2} (\widetilde{f}(z_H^k) - \bar{f}(z_H^k))^2}{4}. \tag{C.12}$$

673 Then we use Lemma C.4 at stage $H$. Let $f = \widetilde{f}_H \in \mathcal{F}_H$, $\widetilde{f} = \bar{f}_H \in \mathcal{F}_H$, $f' = \widehat{f}_{H+1} = 0$. We
674 define:

$$\xi_H^k[f'] := f'(s_{H+1}^k) - V_{H+1}^*(s_{H+1}^k) - [\mathbb{P}_H(f' - V_{H+1}^*)](z_H^k)$$

$$\Delta_H^k[f, \widetilde{f}, f'] := 2\frac{\xi_H^k[f']}{(\widehat{\sigma}_H(z_H^k))^2}\left(f(z_H^k) - \widetilde{f}(z_H^k)\right).$$

675 Therefore, taking a union bound, we have with probability at least $1 - \delta/(4H^2)$, we have

$$\sum_{k\in[K]} 2\frac{\xi_H^k[\widehat{f}_{H+1}]}{(\widehat{\sigma}_H(z_H^k))^2}\left(\widetilde{f}_H(z_H^k) - \bar{f}_H(z_H^k)\right) \leq \left(\frac{4}{3}\iota(\delta)\sqrt{\lambda} + \sqrt{2}\iota(\delta)\right)\|f' - V_{H+1}^*\|_\infty^2$$

$$+ \frac{2}{3}\iota^2(\delta)/\sqrt{\log\mathcal{N}_b} + 30\iota^2(\delta)\|\widehat{f}_{H+1} - V_{H+1}^*\|_\infty^2 + \frac{\sum_{k\in[K]}\frac{1}{(\widehat{\sigma}_H(z_H^k))^2}\left(\widetilde{f}_H(z_H^k) - \bar{f}_H(z_H^k)\right)^2}{4}.$$
(C.13)

676 Combining (C.12) and (C.13), with probability at least $1 - \delta/(2H^2)$, we have

$$2\sum_{k\in[K]}\frac{1}{(\widehat{\sigma}_H(z_H^k))^2}\left(r_H^k + \widehat{f}_{H+1}(s_{H+1}^k) - \bar{f}_H(z_H^k)\right)\left(\widetilde{f}_H(z_H^k) - \bar{f}_H(z_H^k)\right)$$

$$= 2\sum_{k\in[K]}\frac{1}{(\widehat{\sigma}_H(z_H^k))^2}\left(r_H^k + \widehat{f}_{H+1}(s_{H+1}^k) - \mathcal{T}_H\widehat{f}_{H+1}(z_H^k)\right)\left(\widetilde{f}_H(z_H^k) - \bar{f}_H(z_H^k)\right)$$

$$+ 2\sum_{k\in[K]}\frac{1}{(\widehat{\sigma}_H(z_H^k))^2}\left(\mathcal{T}_H\widehat{f}_{H+1}(z_H^k) - \bar{f}_H(z_H^k)\right)\left(\widetilde{f}_H(z_H^k) - \bar{f}_H(z_H^k)\right)$$

$$\leq 2\sum_{k\in[K]}\frac{1}{(\widehat{\sigma}_H(z_H^k))^2}\left(r_H^k + \widehat{f}_{H+1}(s_{H+1}^k) - \mathcal{T}_H\widehat{f}_{H+1}(z_H^k)\right)\left(\widetilde{f}_H(z_H^k) - \bar{f}_H(z_H^k)\right) + 4KL\epsilon$$

$$\leq \frac{4}{3}v(\delta)\sqrt{\lambda} + \sqrt{2}v(\delta) + 30v^2(\delta) + \left(\frac{4}{3}\iota(\delta)\sqrt{\lambda} + \sqrt{2}\iota(\delta)\right)\|f' - V_{H+1}^*\|_\infty^2 + \frac{2}{3}\iota^2(\delta)/\log\mathcal{N}_b$$

$$+ 30\iota^2(\delta)\|\widehat{f}_{H+1} - V_{H+1}^*\|_\infty^2 + 8KL\epsilon + \frac{\sum_{k\in[K]}\frac{1}{(\widehat{\sigma}_H(z_H^k))^2}\left(\bar{f}_H(z_H^k) - \widetilde{f}_H(z_H^k)\right)^2}{2}$$

$$\leq \frac{(\beta_H)^2}{2} + \frac{\sum_{k\in[K]}\frac{1}{(\widehat{\sigma}_H(z_H^k))^2}\left(\bar{f}_H(z_H^k) - \widetilde{f}_H(z_H^k)\right)^2}{2}.$$

677 In the last inequality, we use that fact $\widehat{f}_{H+1} = V_{H+1}^* = 0$ and our choice of $\beta_H$.

$$\beta_H = \sqrt{2\left(\frac{4}{3}v(\delta)\sqrt{\lambda} + \sqrt{2}v(\delta) + 30v^2(\delta) + \frac{2}{3}\iota^2(\delta)/\log\mathcal{N}_b + 8KL\epsilon\right)}$$

$$= \widetilde{O}(\sqrt{\log\mathcal{N}}).$$

678 But conditioned on the event $\mathcal{E}_H$, we have

$$\sum_{k\in[K]}\frac{1}{(\widehat{\sigma}_H(z_H^k))^2}\left(r_H^k + \widehat{f}_{H+1}(s_{H+1}^k) - \bar{f}_H(z_H^k)\right)\left(\widetilde{f}_H(z_H^k) - \bar{f}_H(z_H^k)\right)$$

$$\geq \sum_{k\in[K]}\frac{1}{(\widehat{\sigma}_H(z_H^k))^2}\left(\bar{f}_H(z_H^k) - \widetilde{f}_H(z_H^k)\right)^2$$

$$> \frac{(\beta_H)^2}{2} + \frac{\sum_{k\in[K]}\frac{1}{(\widehat{\sigma}(z_H^k))^2}\left(\bar{f}_H(z_H^k) - \widetilde{f}_H(z_H^k)\right)^2}{2}.$$

679 Here we use (C.11). We finally prove that $\mathbb{P}[\mathcal{E}_H] \geq 1 - \delta/2H^2$.

680 Suppose the event $\mathcal{E}_H$ holds, we can prove the following result.

$$
\begin{aligned}
Q_H^*(s,a) &= \mathcal{T}_H V_{H+1}^*(s,a) \\
&= \mathcal{T}_H \widehat{f}_{H+1}(s,a) \\
&\geq \widetilde{f}_H(s,a) - \left| \mathcal{T}_H \widehat{f}_{H+1}(s,a) - \widetilde{f}_H(s,a) \right| \\
&\geq \widetilde{f}_H(s,a) - \left( \epsilon + |\bar{f}_H(s,a) - \widetilde{f}_H(s,a)| \right) \\
&\geq \widetilde{f}_H(s,a) - b_H(s,a) - \epsilon \\
&= \widehat{f}_H(s,a).
\end{aligned}
$$

681 Here we use the definition of $\widehat{f}_H$ in Algorithm 1 Line 12. Lemma B.6 shows

$$
\sum_{k \in [K]} \frac{1}{(\widehat{\sigma}_h(z_h^k))^2} \left( \bar{f}_h(z_h^k) - \widetilde{f}_h(z_h^k) \right)^2 \leq (\beta_h)^2.
$$

682 Then the fifth inequality follows the bonus oracle assumption (Definition 4.1). Therefore, $V_H^*(s) \geq$
683 $\widehat{f}_H(s)$ for all $s \in \mathcal{S}$.

684 We also have

$$
\begin{aligned}
V_H^*(s) - \widehat{f}_H(s) &= \langle Q_H^*(s,\cdot) - \widehat{f}_H(s,\cdot), \pi^*(\cdot|s) \rangle_{\mathcal{A}} + \langle \widehat{f}_H(s,\cdot), \pi_H^*(\cdot|s) - \widehat{\pi}_H(\cdot|s) \rangle_{\mathcal{A}} \\
&\leq \langle Q_H^*(s,\cdot) - \widehat{f}_H(s,\cdot), \pi^*(\cdot|s) \rangle_{\mathcal{A}} \\
&= \langle \mathcal{T}_H V_H^*(s,\cdot) - \widetilde{f}_H(s,\cdot) + b_H(s,a), \pi^*(\cdot|s) \rangle_{\mathcal{A}} \\
&= \langle \mathcal{T}_H \widehat{f}_{H+1}(s,\cdot) - \widetilde{f}_H(s,\cdot) + b_H(s,a), \pi^*(\cdot|s) \rangle_{\mathcal{A}} \\
&\quad + \langle \mathcal{T}_H V_{H+1}^*(s,\cdot) - \mathcal{T}_H \widehat{f}_{H+1}(s,\cdot), \pi_H^*(\cdot|s) \rangle_{\mathcal{A}} \\
&\leq 2 \langle b_H(s,\cdot), \pi_H^*(\cdot|s) \rangle_{\mathcal{A}} + \epsilon \\
&\leq \widetilde{O} \left( \frac{\sqrt{\log \mathcal{N}} H^2}{\sqrt{K \kappa}} \right).
\end{aligned}
$$

685 Here the second inequality holds because of the definition of $\widehat{\pi}$. The fifth inequality holds due to we
686 the Bellman completeness assumption (Assumption 3.1):

$$
\left| \bar{f}_H(z) - \mathcal{T}_H \widehat{f}_{H+1}(z) \right| \leq \epsilon, \forall z \in \mathcal{S} \times \mathcal{A}.
$$

687 We also use Definition 4.1 and Lemma B.2.

688 Then we do the induction step. Let $R_h = \widetilde{O} \left( \frac{\sqrt{\log \mathcal{N}} H^2}{\sqrt{K \kappa}} \right) \cdot (H - h + 1)$, $\delta_h = h\delta/(4H^2)$. We define
689 another event $\mathcal{E}_h^{\text{ind}}$ for induction.

$$
\mathcal{E}_h^{\text{ind}} = \{ 0 \leq V_h^*(s) - \widehat{f}_h(s) \leq R_h, \forall s \in \mathcal{S} \}.
$$

690 The above analysis shows that $\mathcal{E}_H \subseteq \mathcal{E}_H^{\text{ind}}$ and $\mathbb{P}[\mathcal{E}_H] \geq 1 - 2\delta_H$. Moreover, $\mathbb{P}[\mathcal{E}_H^{\text{ind}}] \geq 1 - 2\delta_H$

691 We conduct the induction in the following way. At stage $h$, if $\mathbb{P}[\mathcal{E}_{h+1}] \geq 1 - 2\delta_{h+1}$ and $\mathbb{P}[\mathcal{E}_{h+1}^{\text{ind}}] \geq$
692 $1 - 2\delta_{h+1}$, we prove that $\mathbb{P}[\mathcal{E}_h] \geq 1 - 2\delta_h$ and $\mathbb{P}[\mathcal{E}_h^{\text{ind}}] \geq 1 - 2\delta_h$.

693 Suppose at stage $h$, $\mathbb{P}[\mathcal{E}_{h+1}] \geq 1 - 2\delta_{h+1}$ and $\mathbb{P}[\mathcal{E}_{h+1}^{\text{ind}}] \geq 1 - 2\delta_{h+1}$. We first use Lemma C.3. Let
694 $f = \widetilde{f}_h \in \mathcal{F}_h$, $f' = \bar{f}_h \in \mathcal{F}_h$. We define

$$
\eta_h^k := V_{h+1}^*(s_{h+1}^k) - [\mathbb{P}_h V_{h+1}^*](z_h^k)
$$

$$
D_h^k[f, f'] := 2 \frac{\eta_h^k}{(\widehat{\sigma}_h(z_h^k))^2} \left( f(z_h^k) - f'(z_h^k) \right).
$$

After taking a union bound, we have with probability at least $1 - \delta/(4H^2)$, the following inequality holds,

$$\sum_{k \in [K]} 2\frac{\eta_h^k}{(\widehat{\sigma}_h(z_h^k))^2} \left( \widetilde{f}(z_h^k) - \bar{f}(z_h^k) \right) \le \frac{4}{3} v(\delta)\sqrt{\lambda} + \sqrt{2}v(\delta) + 30v^2(\delta)$$

$$+ \frac{\sum_{k \in [K]} \frac{1}{(\widehat{\sigma}_h(z_h^k))^2} \left( \widetilde{f}(z_h^k) - \bar{f}(z_h^k) \right)^2}{4}. \qquad \text{(C.14)}$$

Next, we use Lemma C.4 at stage $h$. Let $f = \widetilde{f}_h \in \mathcal{F}_h$, $\widetilde{f} = \bar{f}_h \in \mathcal{F}_h$, $f' = \widehat{f}_{h+1} = \{\widetilde{b}\}_{[0, H-h+1]}$, where $\widetilde{b} = \widehat{f}_h - b_h \in \mathcal{F}_h - \mathcal{W}$. We define:

$$\xi_h^k[f'] := f'(s_{h+1}^k) - V_{h+1}^*(s_{h+1}^k) - \left[ \mathbb{P}_h(f' - V_{h+1}^*) \right](z_h^k)$$

$$\Delta_h^k[f, \widetilde{f}, f'] := 2\frac{\xi_h^k[f']}{(\widehat{\sigma}_h(z_h^k))^2} \left( f(z_h^k) - \widetilde{f}(z_h^k) \right),$$

After taking a union bound, we have with probability at least $1 - \delta/(4H^2)$, we have

$$\sum_{k \in [K]} 2\frac{\xi_h^k[\widehat{f}_{h+1}]}{(\widehat{\sigma}_h(z_h^k))^2} \left( \widetilde{f}_h(z_h^k) - \bar{f}_h(z_h^k) \right) \le \left( \frac{4}{3}\iota(\delta)\sqrt{\lambda} + \sqrt{2}\iota(\delta) \right) \|f' - V_{h+1}^*\|_\infty^2$$

$$+ \frac{2}{3}\iota^2(\delta)/\sqrt{\log \mathcal{N}_b} + 30\iota^2(\delta)\|\widehat{f}_{h+1} - V_{h+1}^*\|_\infty^2 + \frac{\sum_{k \in [K]} \frac{1}{(\widehat{\sigma}_h(z_h^k))^2}(\widetilde{f}_h(z_h^k) - \bar{f}_h(z_h^k))^2}{4}. \qquad \text{(C.15)}$$

Let $U_h$ be the event that (C.14) and (C.15) holds simultaneously. On the event $U_h \cap \mathcal{E}_{h+1}^{\text{ind}}$, which satisfies $\mathbb{P}[U_h \cap \mathcal{E}_{h+1}^{\text{ind}}] \ge 1 - 2\delta_{h+1} - 2\delta/H^2 = 1 - 2\delta_h$, we have

$$2\sum_{k \in [K]} \frac{1}{(\widehat{\sigma}_h(z_h^k))^2} \left( r_h^k + \widehat{f}_{h+1}(s_{h+1}^k) - \bar{f}_h(z_h^k) \right) \left( \widetilde{f}_h(z_h^k) - \bar{f}_h(z_h^k) \right)$$

$$= 2\sum_{k \in [K]} \frac{1}{(\widehat{\sigma}_h(z_h^k))^2} \left( r_h^k + \widehat{f}_{h+1}(s_{h+1}^k) - \mathcal{T}_h\widehat{f}_{h+1}(z_h^k) \right) \left( \widetilde{f}_h(z_h^k) - \bar{f}_h(z_h^k) \right)$$

$$+ 2\sum_{k \in [K]} \frac{1}{(\widehat{\sigma}_h(z_h^k))^2} \left( \mathcal{T}_h\widehat{f}_{h+1}(z_h^k) - \bar{f}_h(z_h^k) \right) \left( \widetilde{f}_h(z_h^k) - \bar{f}_h(z_h^k) \right)$$

$$\le 2\sum_{k \in [K]} \frac{1}{(\widehat{\sigma}_h(z_h^k))^2} \left( r_h^k + \widehat{f}_{h+1}(s_{h+1}^k) - \mathcal{T}_h\widehat{f}_{h+1}(z_h^k) \right) \left( \widetilde{f}_h(z_h^k) - \bar{f}_h(z_h^k) \right) + 4KL\epsilon$$

$$\le \frac{4}{3}v(\delta)\sqrt{\lambda} + \sqrt{2}v(\delta) + 30v^2(\delta) + \left( \frac{4}{3}\iota(\delta)\sqrt{\lambda} + \sqrt{2}\iota(\delta) \right) \|\widehat{f}_{h+1} - V_{h+1}^*\|_\infty^2$$

$$+ \frac{2}{3}\iota^2(\delta)/\log \mathcal{N}_b + 30\iota^2(\delta)\|\widehat{f}_{h+1} - V_{h+1}^*\|_\infty^2 + 8KL\epsilon + \frac{\sum_{k \in [K]} \frac{1}{(\widehat{\sigma}_h(z_h^k))^2} \left( \bar{f}_h(z_h^k) - \widetilde{f}_h(z_h^k) \right)^2}{2}$$

$$\le \frac{(\beta_h)^2}{2} + \frac{\sum_{k \in [K]} \frac{1}{(\widehat{\sigma}(z_h^k))^2} \left( \bar{f}_h(z_h^k) - \widetilde{f}_h(z_h^k) \right)^2}{2},$$

where the third inequality holds because of (C.14) and (C.15). The last inequality holds because on the event of $\mathcal{E}_{h+1}^{\text{ind}}$, $0 \le V_{h+1}^* - \widehat{f}_{h+1} \le R_{h+1} = \widetilde{O}\left( \frac{H^2}{\sqrt{K\kappa}} \right) \cdot (H - h)$ and the choice of $K \ge \widetilde{\Omega}\left( \frac{\iota(\delta)^2 H^6}{\kappa} \right)$. We also use our choice of $\beta_h = \widetilde{O}(\sqrt{\log \mathcal{N}})$.

However, on the event of $\mathcal{E}_h^c$, we have

$$2 \sum_{k \in [K]} \frac{1}{(\widehat{\sigma}(z_h^k))^2} \left( r_h^k + \widehat{f}_{h+1}(s_{h+1}^k) - \bar{f}_h(z_h^k) \right) \left( \widetilde{f}_h(z_h^k) - \bar{f}_h(z_h^k) \right)$$

$$\geq \sum_{k \in [K]} \frac{1}{(\widehat{\sigma}(z_h^k))^2} \left( \bar{f}_h(z_h^k) - \widetilde{f}_h(z_h^k) \right)^2$$

$$> \frac{(\beta_h)^2}{2} + \frac{\sum_{k \in [K]} \frac{1}{(\widehat{\sigma}(z_h^k))^2} \left( \bar{f}_h(z_h^k) - \widetilde{f}_h(z_h^k) \right)^2}{2}.$$

We conclude that $U_h \cap \mathcal{E}_{h+1}^{\mathrm{ind}} \subseteq \mathcal{E}_h$, thus $\mathbb{P}[\mathcal{E}_h] \geq 1 - 2\delta_h$.

Next we prove $\mathbb{P}[\mathcal{E}_h^{\mathrm{ind}}] \geq 1 - 2\delta_h$. Suppose the event $U_h \cap \mathcal{E}_{h+1}^{\mathrm{ind}}$ holds, the above conclusion shows that

$$\sum_{k \in [K]} \frac{1}{(\widehat{\sigma}_h(z_h^k))^2} \left( \bar{f}_h(z_h^k) - \widetilde{f}_h(z_h^k) \right)^2 > (\beta_h)^2.$$

We can prove the following result.

$$\begin{aligned} Q_h^*(s,a) = \mathcal{T}_h V_{h+1}^*(s,a) \\ \geq \mathcal{T}_h \widehat{f}_{h+1}(s,a) \\ \geq \widetilde{f}_h(s,a) - |\mathcal{T}_h \widehat{f}_{h+1}(s,a) - \widetilde{f}_h(s,a)| \\ \geq \widetilde{f}_h(s,a) - (\epsilon + |\bar{f}_h(s,a) - \widetilde{f}_h(s,a)|) \\ \geq \widetilde{f}_h(s,a) - b_h(s,a) \\ = \widehat{f}_h(s,a), \end{aligned}$$

where the second inequality holds because the event $\mathcal{E}_h^{\mathrm{ind}}$ holds. The fourth inequality holds because of the Bellman completeness assumption (Assumption 3.1). From Lemma B.6, we have

$$\sum_{k \in [K]} \frac{1}{(\widehat{\sigma}_h(z_h^k))^2} \left( \bar{f}_h(z_h^k) - \widetilde{f}_h(z_h^k) \right)^2 \leq (\beta_h)^2.$$

Then the fifth inequality holds due to the bonus oracle (Definition 4.1). Therefore, $V_h^*(s) \geq \widehat{f}_h(s)$ for all $s \in \mathcal{S}$.

We also have

$$\begin{aligned} V_h^*(s) - \widehat{f}_h(s) = \langle Q_h^*(s, \cdot) - \widehat{f}_h(s, \cdot), \pi^*(\cdot|s) \rangle_{\mathcal{A}} + \langle \widehat{f}_h(s, \cdot), \pi_h^*(\cdot|s) - \widehat{\pi}_h(\cdot|s) \rangle_{\mathcal{A}} \\ \leq \langle Q_h^*(s, \cdot) - \widehat{f}_h(s, \cdot), \pi^*(\cdot|s) \rangle_{\mathcal{A}} \\ = \langle \mathcal{T}_h V_h^*(s, \cdot) - \widetilde{f}_h(s, \cdot) + b_h(s,a), \pi^*(\cdot|s) \rangle_{\mathcal{A}} \\ = \langle \mathcal{T}_h \widehat{f}_{h+1}(s, \cdot) - \widetilde{f}_h(s, \cdot) + b_h(s,a), \pi^*(\cdot|s) \rangle_{\mathcal{A}} \\ \quad + \langle \mathcal{T}_h V_{h+1}^*(s, \cdot) - \mathcal{T}_h \widehat{f}_{h+1}(s, \cdot), \pi_h^*(\cdot|s) \rangle_{\mathcal{A}} \\ \leq 2 \langle b_h(s, \cdot), \pi_h^*(\cdot, s) \rangle_{\mathcal{A}} + \epsilon + R_{h+1} \\ \leq \widetilde{O} \left( \frac{\sqrt{\log \mathcal{N}} H^2}{\sqrt{K \kappa}} \right) \cdot (H - h + 1) = R_h. \end{aligned}$$

The first equality holds because of our choice of the policy $\widehat{\pi}_h$. In the fifth inequality, we use Assumption 3.1

$$\left\| \bar{f}_h(\cdot, \cdot) - \mathcal{T}_h \widehat{f}_{h+1}(\cdot, \cdot) \right\|_{\infty} \leq \epsilon,$$

and the oracle of bonus function (Definition 4.1) with

$$\sum_{k \in [K]} \frac{1}{(\widehat{\sigma}_h(z_h^k))^2} \left( \bar{f}_h(z_h^k) - \widetilde{f}_H(z_h^k) \right)^2 \leq (\beta_h)^2,$$

718 which holds by Lemma B.6. Therefore, we have $U_h \cap \mathcal{E}_{h+1}^{\text{ind}} \subseteq \mathcal{E}_h^{\text{ind}}$ and $\mathbb{P}[\mathcal{E}_h^{\text{ind}}] \geq 1 - 2\delta_h$. We also
719 use the induction assumption. Thus we complete the proof of induction.

720 Finally, taking the union bound of all the $\mathcal{E}_h$, we get the result that with probability at least $1 - \delta/2$,
721 the event $\cup_{h=1}^{H} \mathcal{E}_h$ holds, i.e for any $h \in [H]$ simultaneously, we have

$$\sum_{k \in [K]} \frac{1}{(\widehat{\sigma}_h(z_h^k))^2} \left( \bar{f}_h(z_h^k) - \widetilde{f}_h(z_h^k) \right)^2 \leq (\beta_h)^2.$$

722 Therefore, we complete the proof of Lemma B.6. $\qquad\square$

# D Auxiliary lemmas

724 **Lemma D.1** (Agarwal et al. 2022). Let $M > 0$, $V > v > 0$ be constants, and
725 $\{x_i\}_{i \in [t]}$ be a stochastic process adapted to a filtration $\{\mathcal{H}_i\}_{i \in [t]}$. Suppose $\mathbb{E}[x_i|\mathcal{H}_{i-1}] = 0$,
726 $|x_i| \leq M$ and $\sum_{i \in [t]} \mathbb{E}[x_i^2|\mathcal{H}_{i-1}] \leq V^2$ almost surely. Then for any $\delta, \epsilon > 0$, let $\iota =$
727 $\sqrt{\log \frac{(2\log(V/v)+2)\cdot(\log(M/m)+2)}{\delta}}$, we have

$$\mathbb{P} \left( \sum_{i \in [t]} x_i > \iota \sqrt{2 \left( 2 \sum_{i \in [t]} \mathbb{E}[x_i^2|\mathcal{H}_{i-1}] + v^2 \right)} + \frac{2}{3}\iota^2 \left( 2 \max_{i \in [t]} |x_i| + m \right) \right) \leq \delta.$$

728 **Lemma D.2** (Regret Decomposition Property, Jin et al. 2021b). Suppose the following inequality
729 holds,

$$|\mathcal{T}_h \widehat{f}_{h+1}(z) - \widetilde{f}_h(z)| \leq b_h(z), \forall z = (s, a) \in \mathcal{S} \times \mathcal{A}, \forall h \in [H],$$

730 the regret of Algorithm 1 can be bounded as

$$V_1^*(s) - V_1^{\widehat{\pi}}(s) \leq 2 \sum_{h=1}^{H} \mathbb{E}_{\pi^*} \left[ b_h(s_h, a_h) \mid s_1 = s \right].$$

731 Here $\mathbb{E}_{\pi^*}$ is with respect to the trajectory induced by $\pi^*$ in the underlying MDP.

732 **Lemma D.3** (Azuma-Hoeffding inequality, Cesa-Bianchi and Lugosi 2006). Let $\{x_i\}_{i=1}^{n}$ be a
733 martingale difference sequence with respect to a filtration $\{\mathcal{G}_i\}$ satisfying $|x_i| \leq M$ for some
734 constant $M$, $x_i$ is $\mathcal{G}_{i+1}$-measurable, $\mathbb{E}[x_i|\mathcal{G}_i] = 0$. Then for any $0 < \delta < 1$, with probability at least
735 $1 - \delta$, we have

$$\sum_{i=1}^{n} x_i \leq M\sqrt{2n\log(1/\delta)}.$$