# OpenReview forum: "Pessimistic Nonlinear Least-Squares Value Iteration for Offline Reinforcement Learning"
_NeurIPS.cc/2023/Conference — Submitted to NeurIPS 2023_

### Official Review · Reviewer_vaFF · 2023-06-21

**Soundness:** 3 good
**Presentation:** 3 good
**Contribution:** 2 fair
**Rating:** 4
**Confidence:** 3

**Summary:**

The paper studies offline RL with non-linear function approximation. The paper is mainly motivated as existing sample complexity guarantees on offline RL algorithms with general function approximation yield suboptimal dependency on the function class complexity, e.g. when the bounds are translated to the linear case. The paper proposes an oracle-efficient algorithm that achieves minimax optimal problem-dependent regret when the bounds are specialized to the linear case. The paper also introduces a new coverage definition.

**Strengths:**

- The paper appears to be technically sound with some new ideas in the algorithm design and formulation of dataset coverage.
- The approach achieve minimax optimal rate in non-linear function approximation, when bounds are converted to linear.

**Weaknesses:**

- The main weakness is that the proposed approach either requires uniform coverage or non-linear bonus oracle. The non-linear bonus oracle is a strong requirement and in effect, simply removes the difficulties related to pessimism in offline RL. On the other hand, the uniform coverage assumption is too strong and thus, it is unfair to compare its efficiency to pessimistic offline RL algorithms.
- A clear comparison to prior work is not presented. In particular, there are multiple axes of comparison, such as dependency on $\epsilon$, dependency on function classes, data coverage requirement, type of oracle, computational efficiency/tractability, realizability assumptions, etc. It is difficult to clearly evaluate the results in this paper without such comparisons. For instance, it will be helpful to have a table as well as translating the bounds of the other algorithms into linear case to see in detail. Additionally, there are several pessimistic offline RL algorithms with general function approximation that only require optimization oracles instead of the more difficult bonus oracle, and no comparison with those papers are presented:

Cheng et al. Adversarially trained actor critic for offline reinforcement learning. In International Conference on Machine Learning (pp. 3852-3878). PMLR

Rashidinejad et al. "Optimal conservative offline rl with general function approximation via augmented lagrangian." arXiv preprint arXiv:2211.00716 (2022).

Ozdaglar et al. Revisiting the Linear-Programming Framework for Offline RL with General Function Approximation. arXiv preprint arXiv:2212.13861

Zhu et al. Importance Weighted Actor-Critic for Optimal Conservative Offline Reinforcement Learning. arXiv preprint arXiv:2301.12714.

**Questions:**

- Is the suboptimality of dependency on the function-class complexity in the bounds of prior algorithms with general function approximation inherent to the algorithm design or a byproduct of analysis?

**Limitations:**

Yes

---

> ### Author Rebuttal · Authors · 2023-08-09
>
> Thank you for your valuable feedback. We address your concerns and questions point-by-point.
>
> **Q1**: Non-linear bonus oracle is a strong requirement. It removes the difficulties related to pessimism in offline RL
>
> **A1**:We would like to clarify that our method does not transfer the primary computational burden to the oracle for identifying the appropriate bonus function. Indeed, in Appendix C of Agarwal et al. (2023), and in Algorithm 2 of Li et al. (2023), they presented an efficient approach to executing this bonus oracle. Therefore, we can provide an efficient implementation of our bonus oracle.
>
> ----
>
> **Q2**: Unfair to compare the efficiency because the assumption is strong and comparison to more prior works.
>
> **A2**: Thank you for raising the issue of fairness in our comparison due to the strength of our assumptions and the need for a broader review of prior works. We will enhance our discussion by making a more extensive comparison. This will include aspects such as the type of algorithm used, the data coverage assumptions made, and the type of oracle utilized. In particular, we have made the following table to facilitate the comparison.
>
> |Algorithm|Algorithm Type |Function Classes|Data Coverage| Types of Oracle |Regret Type  |
> |--------------|-----------------------|--------------------|-------------------|--------|-----|
> | Xie et al. (2021)|Bellman-consistent Pessimism| General|Partial|Optimization on Policy and Function Class| Worst-case
> |CPPO-TV Uehera and Sun (2021)|MLE| General| Partial| Optimization on Policy and Hypothesis Class|Worst-case
> |CORAL Rashidinejad et al. (2022)| MLE|General|Partial|Optimization on Policy and Function Class|Worst-case
> |Reformulated LP Ozdaglar et al. (2023)| Linear Program|General|Partial| Linear Programming|Worst-case|
> | ATAC Cheng et al. (2022) | Actor Critic    | General | Partial |No-regret Policy Optimization & Optimization on the Function class| Worst-case
> |A-Crab Zhu et al. (2023)| Actor Critic    | General | Partial |No-regret Policy Optimization & Optimization on the Function class|Worst-case
> | LinPEVI-ADV+ Xiong et al, (2022) | LSVI-type   | Linear | Uniform | \ | Instance-dependent |
> |PFQL  Yin et al, (2022)|LSVI-type|Differentible|Uniform| Gradient Oracle|Instance-dependent|
> |PNLSVI (Our work)|LSVI-type| General| Uniform| Bonus Oracle& Optimization on the Function class|Instance-dependent
>
> ----
>
> **Q3**: Is the suboptimality of dependency on the function-class complexity in the bounds of prior algorithms with general function approximation inherent to the algorithm design or a byproduct of analysis?
>
> **A3**：We believe that it is inherent to algorithm design. Without using the variance information of the value function and the Bernstein-type concentration inequality to construct the bonus/confidence set, we don’t think they can achieve optimal dependency on the function-class complexity.
>
> ----
>
> [1] Agarwal et al. (2023). Vo q l: Towards optimal regret in model-free rl with nonlinear function approximation. COLT
>
> [2] Li, et al. (2023) Low-switching policy gradient with exploration via online sensitivity sampling. arXiv preprint
>
> [3] Xie et al. (2021). Bellman-consistent pessimism for offline reinforcement learning. In NeurIPS
>
> [4] Uehara and Sun (2021). Pessimistic model-based offline reinforcement learning under partial coverage. In ICLR.
>
> [5] Rashidinejad et al. (2022). Optimal conservative offline rl with general function approximation via augmented lagrangian. In ICLR.
>
> [6] Ozdaglar et al. (2023). Revisiting the linear-programming framework for offline rl with general function approximation. In ICML.
>
> [7] Cheng et al.(2022). Adversarially trained actor critic for offline reinforcement learning. In ICML
>
> [8] Zhu et al.(2023). Importance weighted actor-critic for optimal conservative offline reinforcement learning. arXiv preprint
>
> [9] Xiong et al. (2023). Nearly minimax optimal offline reinforcement learning with linear function approximation: Single-agent mdp and markov game. In ICLR
>
> [10] Yin et al.(2022). Offline reinforcement learning with differentiable function approximation is provably efficient. In ICLR

---

### Official Review · Reviewer_17B8 · 2023-06-27

**Soundness:** 3 good
**Presentation:** 3 good
**Contribution:** 2 fair
**Rating:** 5
**Confidence:** 5

**Summary:**

This paper considers variance-weighted least-squared regression for offline RL with general function approximation. Under a uniform data coverage assumption, they show that the proposed algorithm obtains a sub-optimality bound that scales with the $D^2$-divergence of the offline data set, the positive lower-bounded constant of the uniform data coverage, and the complexity of the function class. Their bound obtains the right order when realized in the linear case.

**Strengths:**

- clear presentation (though some parts can be improved further -- see Weaknesses)
- the obtained result is new and relevant to the offline RL community


**Weaknesses:**

- The main weakness is that the uniform data coverage assumption is very strong. In the linear case, this assumption is equivalent to that the behavior policy is exploratory overall dimensions of the linear feature. A question for the authors is that in such a case, why would we even need pessimism?  Pessimism is used when the data coverage is partial thus we become pessimistic about uncertain actions. But when the coverage is uniform, it can eliminate the need for pessimism and we can simply use greedy algorithms. I understand that without such a uniform data coverage assumption, it seems difficult to get a reliable estimation of the variance of the transition kernel and it would be interesting to get rid of this assumption. But if we could not get rid of it yet, the very least expectation is that we need to explain this assumption further, especially regarding where pessimism is really needed with this assumption.


- Writing can be improved further. For example, the $D^2$-divergence and the definition of the bonus function (Def 4) can be explained and motivated further. The current presentation of these concepts are not very helpful

- Some claims might be potentially misleading. It's not comfortable to view the proposed algorithm as computationally efficient even in the oracle sense. Specifically, the construction of the bonus function in Definition 4.1 is far from being computationally efficient since it is essentially a constrained optimization over the version space. That said, it is nowhere more computationally efficient than version-space-based algorithms such as the "Bellman-consistent pessimism" of Xie et al.

- Though the main result is new, it appears expected given the already-developed machinery in Argawal et al. 2022 and Xiong et al. 2022. What are the technical challenges in the current problem that the existing techniques cannot resolve?

- Some minor: PNLSVI is never introduced before used

**Questions:**

Please see the Weaknesses section

**Limitations:**

Yes

---

> ### Author Rebuttal · Authors · 2023-08-09
>
> Thank you for your valuable feedback. We address your concerns and questions point-by-point.
>
> **Q1**: Uniform data coverage assumption is very strong, in such a case, why would we even need pessimism?
>
> **A1**: We agree that pessimism does not require the uniform data coverage assumption. However, since we want to leverage the variance information of the value function, we made this assumption. Note that our assumption can be reduced to that in Xiong et al. (2023) and Yin et al.(2023) in the linear function approximation setting, because they also need to leverage the variance information. How to relax this assumption to partial coverage or single concentrability assumption will be an interesting work. We will study it in the future.
>
> ----
>
> **Q2**: The $D^2$-divergence and the definition of the bonus function (Def 4) can be explained and motivated further.
>
> **A2**: Thank you for your suggestion. In our revision, we will provide a detailed explanation as follows.
>
> The $D^2$ divergence, defined as $D_{\mathcal F_h}^2(z;\mathcal D_h; \sigma_h) = \sup_{f_1,f_2 \in \mathcal F_h}\frac{(f_1(z)-f_2(z))^2}{\sum_{k \in [K]}\frac{1}{(\sigma_h(z_h^k))^2}(f_1(z_h^k)-f_2(z_h^k))^2 + \lambda}$, is a measure introduced to quantify the disparity of a given point $z=(s,a)$ from the historical dataset $\mathcal D_h$​. It signifies the extent to which the behavior of functions within the function class can deviate at the point $z=(s,a)$, based on their difference in the historical dataset.  It can be viewed as the generalization of elliptical norm $\|\phi(s,a)\|_ {\Sigma_h^{-1}}$ in linear case, where $\Sigma_h$ is defined as $\sum_{k \in [K]}\phi(s_h^k,a_h^k)\phi(s_h^k,a_h^k)^\top + \lambda \mathbf{I}$.
>
> By employing the pessimism principle, we hope to design our policy based on the worst-case guarantee on its expected return. One significant challenge arises when trying to prevent overestimation. This necessitates us to quantify functional uncertainty, denoted by $\max_{f_1,f_2 \in \mathcal{F}}|f_1(s_h,a_h)-f_2(s_h,a_h)|$. But the inherent complexity of this uncertainty bonus is huge, concerning an optimization of functions. Therefore, we assume the existence of a bonus oracle with reduced complexity (Def 4.1). This bonus oracle can indeed be efficiently implemented using the method proposed in Appendix C of Agarwal et al. (2023) and Algorithm 2 of Li et al. (2023).
>
> ----
>
> **Q3**: Computationally efficiency: bonus function is not efficient.
>
> **A3**: In line 55, we said "Our algorithm is oracle-efficient, i.e., it is computationally efficient when there exists an efficient regression oracle and bonus oracle for the function class".
> We would like to clarify that the bonus oracle can indeed be efficiently implemented using the method proposed in Appendix C of Agarwal et al. (2023) and Algorithm 2 of Li et al. (2023). They offer an effective and efficient algorithm for implementing the bonus oracle.
>
> Nevertheless, we acknowledge your concerns and are open to modifying our statement for the sake of clarity. We propose revising it to state, "Our algorithm is computationally tractable given the bonus oracle”.
>
> ----
>
> **Q4**: PNLSVI is never introduced before used
>
> **A4**: Its full name is Pessimistic Nonlinear Least-Square Value Iteration (PNLSVI). We will provide its full name in the abstract and the introduction.
>
> ----
>
> [1] Agarwal et al. (2023). Vo q l: Towards optimal regret in model-free rl with nonlinear function approximation. In COLT
>
> [2] Li et al. Low-switching policy gradient with exploration via online sensitivity sampling. arXiv preprint
>
> [3] Xiong et al. (2023). Nearly minimax optimal offline reinforcement learning with linear function approximation: Single-agent mdp and markov game. In ICLR
>
> [4] Yin et al. (2022). Offline reinforcement learning with differentiable function approximation is provably efficient. In ICLR

---

> > ### Comment · Reviewer_17B8 · 2023-08-17
> >
> > I thank the authors for the response. I would like to keep my initial evaluation.

---

> > > ### Author Response · Authors · 2023-08-17
> > >
> > > Thank you for your positive feedback.

---

### Official Review · Reviewer_gXdR · 2023-07-06

**Soundness:** 1 poor
**Presentation:** 2 fair
**Contribution:** 1 poor
**Rating:** 3
**Confidence:** 3

**Summary:**

This paper proposes a pessimistic nonlinear least-squares value iteration algorithm to tackle the offline reinforcement learning problem. The main motivation of the paper is to propose an algorithm that are both computationally efficient and minimax optimal w.r.t. the complexity of nonlinear function class. The proposed pessimism-based algorithm strictly generalizes the existing pessimism-based algorithms for both linear and differentiable function approximation and is oracle efficient. Also, the proposed algorithm is proven to be optimal w.r.t. the function class complexity, closing the gap originated from the previous work on differentiable function approximation.

**Strengths:**

1) The proposed algorithm is proven to be optimal w.r.t. the complexity of nonlinear function class, closing the gap from the previous work on the differentiable function class and generalizes it to the wider nonlinear function class.
 2) The proposed algorithm is computationally efficient if there exist the efficient oracles for both regression minimization and bonus function optimization/searching.

**Weaknesses:**

1) The paper's presentation needs some work. For example, the terminology definition is not consistent. The D^2 divergence definition in Definition 3.2 is not consistent with the later terminology of D_F in line 239. The language itself needs some work too. For example, lots of places where it needs 'an', but 'a' is used and vice versa. Please define RL before using it in the abstract. There are also some ambiguities in the definitions that needs clarification in the Question section.
2) The paper's claimed contribution is a bit exaggerated. Although the proposed algorithm does not need the computationally heavy optimization as previous works in planning phase, it transfers the main computation burden to the Oracle to find the satisfied bonus function, which seems to be a very time-consuming task. It also applies to the claim of being the first statistically optimal algorithm for nonlinear offline RL. Being able to get optimal result in the reduced linear function class does not necessarily mean it's optimal in the broader nonlinear class.
3) Although the considered class is the nonlinear one and general than the previously considered linear or differentiable class, the techniques used in the analysis are nothing new in my opinion, except re-defining the metrics in the nonlinear function class and connect the results together along with additional assumptions.
Overall, I think the paper is well motivated, but given the presentation and the insignificant contribution, it's not ready to be published.

**Questions:**

1) In line 137, the paper defines the Bellman operator for the function f: S->R. Then why in the operator, it takes both state and action as input?
2) In line 141, should we replace the V with Q?
3) In line 172, the generalized definition to offline setting, If D_h corresponds to the observations collected up to stage h in the MDP, what is z_h^k? Is it only one (state,action) pair or the observations collected up to stage h?
4) Line 214, should the function be all \hat{f^{\prime}}?

---

> ### Author Rebuttal · Authors · 2023-08-09
>
> Thank you for your valuable feedback. We address your concerns and questions point-by-point.
>
> **Q1**: The presentation needs some work.
>
> **A1**: Thanks for your feedback. We will address these problems one by one.
>
> - The consistency of the $D^2$ definition:
> Our current definition in Section 3.2 pertains to the online setting, whereas in line 172, we present the $D^2$ definition suited to the offline setting. This definition is intended to be consistent with the notation we adopt later in the paper. To emphasize our new definition, we will clarify it in the revision, and we will also include a remark comparing it with the online setting to ensure a comprehensive understanding. We apologize for any confusion our initial presentation may have caused and thank you for pointing this out.
>
>
> - Writing errors and notation issues:
> We will thoroughly proofread the entire document to rectify any grammatical mistakes and improve the clarity of the text.
> Regarding your comment on line 172, we introduced the shorthand notation where $z = (s, a)$ and $z_h^k = (s_h^k,a_h^k)$. We will define this notation in the revision.
> Further, we are grateful for your observation on line 214. It appears to be a typo and we will promptly correct it.
>
> - Potential ambiguity of the use of the Bellman operator:
> We define the Bellman operator on line 137, where it is intended to accept both state and action as inputs according to our definition. To mitigate any possible ambiguity, we will incorporate brackets [] to explicitly denote that the operator is being applied to a function f which takes only state as inputs and then get a function taking state and action as input, i.e., $\[ \mathcal T_h f \] (s_h,a_h)$.   $= E_{s_{h+1} \sim \mathbb P_h (\cdot|s_h,a_h)}$   $[r_h(s_h,a_h) + f(s_{h+1})]$.
> In relation to line 141, the function $V$ is used in place of $f$ from the definition. Therefore, upon application of the Bellman operator, it is intended to take both state and action as inputs. We will also use $\[ \mathcal{T}_h V_{h+1} ^{\pi}\](s_h,a_h)$ to avoid confusion.
>
> ----
>
>
> **Q2**: contribution is exaggerated
>
> **A2**: We will revise this part to present a more accurate representation without making excessive claims.
>
>
> We appreciate your keen observation that the optimal result in the reduced linear function class does not necessarily imply optimality in the broader nonlinear class. Yet this can still indicate that our data-dependent regret bound is reasonably tight, especially the dependency on the function-class complexity.
>
> We would also like to clarify that our algorithm does not transfer the primary computational burden to the oracle for identifying the appropriate bonus function. Indeed, in Appendix C of Agarwal et al. (2023), and in Algorithm 2 of Li et al. (2023), they presented an efficient approach to executing this bonus oracle. Therefore, we can provide an efficient implementation of our bonus oracle. So our algorithm is indeed computationally-efficient.
>
>
> ----
>
> [1] Agarwal et al. (2023). Vo q l: Towards optimal regret in model-free rl with nonlinear function approximation. In COLT
>
> [2] Li et al. (2023) Low-switching policy gradient with exploration via online sensitivity sampling. arXiv preprint

---

> > ### Comment · Reviewer_gXdR · 2023-08-18
> >
> > Thank you for taking time to reply my review. I did check the Algorithm 2 of Li et al. (2023) mentioned in the rebuttal, but the computation concern is still there. As a result, I will keep my score unchanged.

---

> > > ### Author Response · Authors · 2023-08-18
> > >
> > > Dear reviewer gXdR,
> > >
> > > Thank you for your reply. In order to fully understand and address your concern and improve our current work, we'd like to kindly request further clarification. Could you please explain what the computational burden is in this algorithm? In our opinion, this is no less efficient than the regression oracle used in other papers (i.e., the computational overhead for both the regression oracle and the bonus oracle are similar). Thanks.
> > >
> > > Best,
> > >
> > > Authors

---

### Author Rebuttal · Authors · 2023-08-09

Dear reviewers,

Based on the questions of reviewer gXdR and reviewer 17B8 on technical challenges, we would like to emphasize the difficulty of this problem being studied and our novel techniques to tackle it. Firstly, the variance information in the context of Least-Squares Value Iteration (LSVI) type algorithms is crucial to obtain better statistical efficiency. Under linear function approximation, the value function is always a linear function with respect to the feature mapping $\phi$, and it is straightforward to approximate the value function by directly estimating the underlying linear parameter $\theta$. However, the situation becomes more complex with nonlinear function approximation since the value function is no longer a linear function. In the nonlinear function approximation in offline RL, we propose novel constructions and concentration inequalities. In detail, we approximate the value function and the square of the value function  by nonlinear least-squares regression

$\tilde f_h' = \underset{f_h \in \mathcal F_h}{\operatorname{argmin}} \sum_{k \in [K]}  (f_h(\bar s_h^k,\bar a_h^k)-\bar r_h^k - \hat f_{h+1}'(\bar s_{h+1}^{k}))^2$ and

$ \tilde g_h'= \underset{g_h \in \mathcal F_h}{\operatorname{argmin}} $$ \sum_{k \in [K]} (g_h(\bar s_h^k,\bar a_h^k)-(\bar r_h^{k} + \hat f_{h+1}'(\bar s_{h+1}^{k}))^2)^2$ (Algorithm 1, Lines 3-4).

We construct the following confidence interval for nonlinear function approximation
$\sum_{k \in [K]}(\bar f_h'(\bar z_h^k) - \tilde f_h' (\bar z_h^k))^2 \leq (\beta_{1,h}')^2$ and   $\sum_{k \in [K]}\left(\bar g_h'(\bar z_h^k) - \tilde g_h' (\bar z_h^k)\right)^2 \leq (\beta_{2,h}')^2$ (Lemmas 6.1 and 6.2). With the help of the concentration properties, we could estimate the variance information and further provide a variance-dependent estimation for the value function $\tilde f_h = \underset{f_h \in \mathcal F_h}{\operatorname{argmin}} \sum_{k \in [K]} \frac{1}{\hat \sigma_h^2(s_h^k,a_h^k)}(f_h(s_h^k,a_h^k)-r_h^k - \hat f_{h+1}(s_{h+1}^k))^2$ (Algorithm 1, Line 10).

Additionally, we address the issue of reference-advantage decomposition. This method is an effective tool for overcoming the challenge posed by the additional error from uniform concentration over the entire function class $F_h$. Xiong et al. (2023) employed an estimate of the Bellman operator for decomposition, but in the nonlinear setting, a direct counterpart does not exist. Prior works, such as Yin et al. (2022), were unsuccessful in adapting this technique to the nonlinear function class, resulting in a suboptimal dependency on the complexity of the function class. In comparison, we decompose the bellman error to reference uncertainty $r_h(s,a) + f_{h+1}^*(s,a) - \[\mathcal T_h f_{h+1}^*\](s,a)$ and Advantage uncertainty $\hat f_{h+1}(s,a) - f_{h+1}^*(s,a) -( \[\mathbb P_h \hat f_{h+1}](s,a) - \[\mathbb P_h f_{h+1}^*\](s,a))$ (Line 313). For the reference uncertainty, the optimal value function $f^*_{h+1}$ is fixed and not related to the pre-collected dataset, which circumvents additional uniform concentration over the whole function class and avoids the dependence on the function class size. For the advantage uncertainty, it is worth noticing that the distance between the estimated function $\hat{f}'_{h+1}$ and the optimal value function $f_h^*$ is decreased as $O(1/\sqrt{K\kappa})$. Though, we still need to maintain the uniform convergence guarantee, the advantage uncertainty is dominated by the reference uncertainty when the number of episodes $K$ is large enough. (Line 315-318)


To our knowledge, we are the first to utilize this method in a nonlinear function approximation and prove an optimal dependency on the complexity of the function class when it is specialized to the linear case.

----

Additionally, based on the feedback of reviewer vaFF, we have made a table to provide a comprehensive comparison of different algorithms. Please find the table in the uploaded pdf file.

----

[1] Xiong et al. (2023). Nearly minimax optimal offline reinforcement learning with linear function approximation: Single-agent mdp and markov game. In ICLR

[2] Yin et al. (2022). Offline reinforcement learning with differentiable function approximation is provably efficient. In ICLR

---

### Decision · Program_Chairs · 2023-09-21

**Decision:**

Reject

**Comment:**

I really appreciate the authors for conducting the rebuttal and the following discussions. Overall this paper conducted nice result on the offline RL with nonlinear LSVI. The overall result is strong, however, the connection with other paper e.g. [Yin et al.] need to be further clear  discussed as the regression scheme and variance reweighting share a lot of similarity. Also the suggestions mentioned by other reviewers. Although the overall result is good, due to the competitiveness of NeurIPS, this paper needs careful revisions before getting accepted. I recommend the paper be rejected. But I do believe it will be accepted by top ML conferences in the future.